# Processes influencing lower stratospheric water vapour in monsoon anticyclones: insights from Lagrangian modeling

Nuria Pilar Plaza[1], Aurélien Podglajen[2], Cristina Peña-Ortiz[1], and Felix Ploeger[3,4]

[1]Área de Física de la Tierra. Departamento de Sistemas Físicos, Químicos y Naturales. Universidad Pablo de Olavide, Sevilla, Spain
[2]Laboratoire de Météorologie Dynamique (LMD/IPSL), École polytechnique, Institut polytechnique de Paris, Sorbonne Université, École normale supérieure, PSL Research University, CNRS, Paris, France
[3]Institute of Climate Research. Forschungszentrum Jülich, Jülich, Germany
[4]Institute for Atmospheric and Environmental Research, University of Wuppertal, Wuppertal, Germany

**Correspondence:** Nuria Pilar Plaza (npplamar@upo.es)

**Abstract.** We investigate the influence of different chemical and physical processes on the water vapour distribution in the lower stratosphere (LS), in particular in the Asian and North-American monsoon anticyclones (AMA and NAMA, respectively). Specifically, we use the chemistry transport model CLaMS to analyze the effects of large-scale temperatures, methane oxidation, ice microphysics, and small-scale atmospheric mixing processes in different model experiments. All these processes hydrate the LS and, particularly, the AMA. While ice microphysics has the largest global moistening impact, it is small-scale mixing which dominates the specific signature in the AMA in the model experiments. In particular, the small-scale mixing parameterization strongly contributes to the water vapour transport to this region and improves the simulation of the intra-seasonal variability, resulting in a better agreement with MLS observations. Although none of our experiments reproduces the spatial pattern of the NAMA as seen in MLS observations, they all exhibit a realistic annual cycle and intra-seasonal variability, which are mainly controlled by large-scale temperatures. We further analyse the sensitivity of these results to the domain-filling trajectory set-up, here-called Lagrangian Trajectory Filling (LTF). Compared with MLS observations and with a multiyear reference simulation using the full-blown chemistry transport model version of CLaMS, we find that the LTF schemes result in a drier global LS and in a weaker water vapour signal over the monsoon regions, which is likely related to the specification of the lower boundary condition. Overall, our results emphasize the importance of subgrid-scale mixing and multiple transport pathways from the troposphere in representing water vapour in the AMA.

## 1 Introduction

Water vapour in the upper troposphere-lower stratosphere (UTLS) is one of the most important chemical species because of its impact on the global radiative budget (Solomon et al., 2010; Riese et al., 2012). Its distribution depends on the strength of the Brewer–Dobson circulation, the quasi-horizontal isentropic transport between tropical and high latitudes and the convective

activity that enhances the cross-isentropic transport (Fueglistaler and Haynes, 2005; Diallo et al., 2018; Poshyvailo et al., 2018). The Brewer–Dobson circulation lifts up moist air from the troposphere into the deep stratosphere through the Tropical Tropopause Layer (TTL). While crossing the TTL, air masses encounter the cold temperatures of the tropopause, the so-called Cold Point Tropopause (CPT) (Fueglistaler et al., 2005), resulting in ice formation, sedimentation and dehydration of the ascending air parcels. A number of studies have shown that, at first order, water vapour entering the stratosphere responds to the variability in CPT temperature (e.g. Mote et al., 1996; Fueglistaler and Haynes, 2005; Randel and Park, 2019). In particular, the pronounced annual cycle of tropical tropopause temperature is responsible for dry and wet anomalies which propagate upward in the tropical lower stratosphere, forming the water vapour tape recorder (Mote et al., 1996). However, the control of water vapour anomalies by the CPT weakens during boreal summer, when maxima of water vapour in the UTLS are found over the Asian and North American monsoon regions (Randel and Park, 2019). This raises the question of the importance of the monsoon systems as a secondary pathway to transport water vapour into the LS.

Monsoon circulations appear as a dynamical response to diabatic heating released by persistent convection over regions close to the equator (Gill, 1980). In the case of the Asian Monsoon, convection has its climatological center over the Bay of Bengal, and generates the AMA, a strong planetary-scale anticyclone in the UTLS which is the most dominant feature in the global atmosphere during boreal summer (Hoskins and Rodwell, 1995). The rapid vertical transport in the inner core of the monsoon pumps up moist air masses from the troposphere directly into the UTLS. There, the strong anticyclonic winds of the monsoon circulation behave as a transport barrier (Ploeger et al., 2015) that isolate the air masses from outer regions, keeping the air with high water vapour content (and similar for other trace gases with tropospheric sources) confined (Randel and Park, 2006; Park et al., 2007; Randel et al., 2010; Santee et al., 2017). At this height, air masses slowly ascend through the cold tropopause, where they further dehydrate (Park et al., 2007). However, the mechanism of simple large-scale temperature control alone might not be sufficient to explain water vapour distributions by itself as air masses in the Asian monsoon UTLS are generally about 20–50% supersaturated (e.g., Krämer et al., 2020). In the case of the NAMA, there is less understanding of the water vapour signal observed, which is much stronger than for air under purely saturated conditions (Gettelman et al., 2004). However, as the anticyclonic circulation in the UTLS of the North American monsoon is much weaker, the sensitivity to processes also present in the Asian Monsoon, such as convection, is different (Dessler and Sherwood, 2004; Randel et al., 2012). Thus, besides tropical cold-point temperatures, multiple other factors influence the transport of water vapour to the LS.

The impact of convection has been the focus of several studies but remains controversial. While Dessler and Sherwood (2004); Dessler et al. (2007); Ueyama et al. (2018) and Gettelman et al. (2002a) found that convection, and especially over-shooting events, increase the LS water vapour signal over the monsoon regions, other studies emphasized that the main role of convection is related to changes in diabatic heating rates and, hence, the dynamical structure of the region (Gettelman et al., 2002b; Park et al., 2007; Schoeberl et al., 2013; Randel et al., 2015; Zhang et al., 2016; Kim et al., 2018). Randel et al. (2015) found that stronger convection leads to relatively cold temperatures in the subtropical LS, which they identified as a key region controlling large-scale dehydration within the anticyclonic monsoonal circulation, giving rise to a drier stratosphere. Therefore, it is not clear whether the main role of convection is to moisten the LS through overshooting events or to dehydrate it by decreasing the tropopause temperatures.

Furthermore, Ueyama et al. (2018) concluded from Lagrangian experiments that convective hydration is necessary to explain the water vapour signal over monsoon regions. On the other hand, James et al. (2008) found that process to be of second order and other studies achieved realistic H2O distributions without any convective scheme (Schoeberl et al., 2013; Ploeger et al., 2013; Poshyvailo et al., 2018). Thus, this apparent disagreement not only highlights the problem to understand the role of convection in LS water vapour simulations, but also the impact that the configuration of a model experiment might have on the LS water vapour distribution. Schoeberl et al. (2013) uses the model developed by Schoeberl and Dessler (2011) based on the domain-filling Lagrangian technique. This approach is based on the philosophy that LS water vapour depends on the processes acting in the upper troposphere and in the tropopause region and therefore assumes only a minor role of the lower to mid tropospheric water vapour distribution and the specific model set-up of air parcel release at locations close but below the tropopause. This approach has been successfully applied to answer many questions related to the water vapour distribution in the TTL (Schoeberl and Dessler, 2011; Schoeberl et al., 2012, 2013, 2014; Zhang et al., 2016; Schoeberl et al., 2018, 2019; Wang et al., 2019), but it has never been compared in detail to consistent models covering the whole troposphere as well.

Another relevant process to the water vapour budget is ice microphysics (in particular, sedimentation and detrainment) in the UTLS related to the formation of cirrus clouds. Ice could be convectively lofted (Corti et al., 2008; Dessler et al., 2007, 2016; Ueyama et al., 2018; Wang and Dessler, 2012; Schoeberl et al., 2019) or in situ formed (Wang and Dessler, 2012; Ploeger et al., 2013; Krämer et al., 2020). In the first case, it is not clear whether evaporation of ice injected into the LS by overshoots leads to a moistening of the LS (Corti et al., 2008; Wang and Dessler, 2012) or not (Ueyama et al., 2018). In the second case, cirrus clouds form in cold regions of the UTLS (Gettelman et al., 2002a), decreasing the water vapour present. However, depending on their properties, such as their thickness, cirrus clouds could lead to a warming of these regions (Krämer et al., 2020). This agrees with Ploeger et al. (2013), which showed from model simulations that evaporation of ice in the UTLS increases the water vapour everywhere, including the Asian Monsoon region. However, the relative role of this process in contrast with other mechanisms not only on net water vapour in monsoon anticyclones but also on its variability, has not been fully assessed yet.

Turbulence and the associated small-scale mixing results in diffusivity in the UTLS, which affects the transport of trace gas constituents, including water vapour, into the LS (Podglajen et al., 2017). Konopka et al. (2007) showed that a parameterization of the small-scale mixing between nearby air masses based on the strain- and shear- induced deformation of the large-scale flow led to an enhancement of cross-tropopause transport in the monsoon regions and in particular in the Asian Monsoon. This mechanism has been invoked to explain observed tracer distributions in the UTLS (Pan et al., 2006). As flow deformation is commonly found in the vicinity of the subtropical jet stream, which is very close to the tropopause, air masses tend to mix in these regions. As a consequence, air masses reach the LS with higher water vapour content, avoiding in some cases the coldest temperatures of the tropopause (Poshyvailo et al., 2018). However, as reported by Poshyvailo et al. (2018) and Riese et al. (2012), the final impact of mixing on water vapour largely depends on the mixing strength predefined in their simulations and is thus highly uncertain.

At mid-stratospheric levels, methane oxidation acts as a source of water vapour. Through the downwelling branch of the Brewer–Dobson circulation, these moistened air masses are transported into the LS, and partly are further recirculated into the

tropics (Ploeger et al., 2013). Despite the fact that this horizontal transport is not as strong as in the opposite direction, it has a non-negligible impact on the monsoon regions.

In this study, we use the Chemical Lagrangian Model of the Stratosphere (CLaMS) (McKenna et al., 2002b, a; Konopka et al., 2004) with the aim of describing and quantifying the contributions of the different physical processes to the water vapour distribution in the lower stratosphere and particularly over the Asian and American monsoons. For this purpose, we have performed five experiments to analyse the role of each of the following processes: large-scale temperatures, methane chemistry, ice microphysics (including effects of ice sedimentation and nucleation barrier), small-scale mixing processes, and in particular vertical tropospheric mixing (likely related to convection). Furthermore, we also assess the sensitivity of the LS water vapour signal to the domain-filling technique developed by Schoeberl and Dessler (2011). This approach has been widely used in the recent past (i.e Zhang et al., 2016; Schoeberl et al., 2018; Wang et al., 2019), but the effects of this set-up on simulated water vapour distributions have not been studied in detail yet. To shed more light on the related effects, we developed a model version of CLaMS analogous to this forward trajectory domain-filling approach, configured all the sensitivity experiments based on this model set up and compared them with a multi-decadal full-blown chemistry transport model CLaMS simulation as used in Konopka et al. (2004); Diallo et al. (2018); Tao et al. (2019). Besides, we used satellite observations from Aura MLS to assess the reliability of each simulation.

The remainder of the paper is organized as follows. In Section 2, we present the domain-filling technique, the data and the configuration of the different experiments. In Section 3, we evaluate the different experiments in simulating LS water vapour and how they capture the variability of the water vapour signal over the Asian and North American monsoon regions. Finally, in Section 4, we discuss the relevance of the processes to simulate the water vapour signal and also the differences in water vapour found using the domain-filling technique and the standard version of CLaMS. It should be noted that we do not aim to provide a most realistic model here but rather carry out simplified sensitivity experiments to present estimates of the effects of various processes to be potentially included in models simulating water vapour in the monsoon UTLS.

## 2  Data and methodology

### 2.1  The CLaMS model

To evaluate the sensitivity of lower stratospheric water vapour over monsoon regions to different physical processes, we use the Chemical Lagrangian transport model CLaMS (McKenna et al., 2002b; Konopka et al., 2004). This model simulates the three-dimensional trajectories of an ensemble of air parcels forward in time, as well as the changes in the chemical composition of the air parcels along them. CLaMS has a modular structure that allows different parameterizations or new configurations to be easily implemented. Thus, the sensitivity of the water vapour distribution to each parameterization can be studied easily by switching them on and off.

The CLaMS model has been widely used to study the distribution of several tracers in the stratosphere (Riese et al., 2012), including recent studies on water vapour in the lower stratosphere (Tao et al., 2019; Poshyvailo et al., 2018). Previous studies have shown that the model properly simulates the variability of the stratospheric water vapour (Diallo et al., 2018; Tao et al.,

2019) and the water vapour distribution over monsoon regions during boreal summer (Poshyvailo et al., 2018), highlighting the efficiency of these regions to transport air masses and water vapour into the TTL (Ploeger et al., 2017; Nützel et al., 2019; Yan et al., 2019).

All experiments performed for the present study use 6-hourly winds and temperature from the European Centre of Medium-range Weather Forecast (ECMWF) ERA-Interim reanalysis (Dee et al., 2011). The model uses a vertical hybrid coordinate that follows the orography with a $\sigma$-coordinate at the ground that transforms into potential temperature in the upper troposphere. Above $\sigma$=0.3 (about 300 hPa in regions without strong orography), the vertical coordinate is purely isentropic. Cross-isentropic transport is simulated using total diabatic heating rates (considering all-sky radiation, latent heat release, diffusive and turbulent heat transport as detailed by Fueglistaler et al. (2009)) from ERA-interim forecast data, as described in Ploeger et al. (2010). For the sake of the analysis, simulated water vapour content of air parcels is daily gridded into maps with bin size 5°-longitude x 2°-latitude at a given pressure or potential temperature level with a thickness of 10 hPa or K, respectively. Hence, daily distributions of water vapour at 100 hPa are result of averaging air parcels found between 105 and 95 hPa.

## 2.2  Domain filling set up

To create a common framework between previous studies focusing on the simulation of stratospheric water vapour (Schoeberl et al., 2013; Zhang et al., 2016; Wang et al., 2019) and our CLaMS sensitivity experiments, we have implemented the forward domain-filling technique, here referred as Lagrangian Trajectory Filling (LTF), into CLaMS. This set-up, introduced by Schoeberl and Dessler (2011), has been widely used to study different properties of water vapour in the stratosphere and UTLS region (Schoeberl et al., 2012, 2013, 2014; Dessler et al., 2014; Zhang et al., 2016; Ye et al., 2018; Schoeberl et al., 2018, 2019). In this approach, air parcels are continuously launched at a given level below the tropopause and their trajectories are calculated forward in time until they leave the domain of interest, here bounded by the surfaces $p = 250 \, \mathrm{hPa}$ (lower boundary) and $\theta = 1800$ K. After a spin-up time during which the number of tracked air parcels increases, an equilibrium state is reached in which the release of new air parcels balances removal at the boundaries. At that point, due to the structure of the large-scale stratospheric circulation, the whole domain is filled with air parcels.

Our set-up closely follows that of Schoeberl and Dessler (2011). Once a day (at 12 UTC), air parcels are released on a regular 5°-longitude x 2°-latitude grid spanning the 60°S–60°N latitudinal band. We initialize on the $\theta = 360$ K surface, which is, on average, above the level of zero radiative heating (LZRH) (Gettelman et al., 2002a) but below the tropical tropopause (∼375–380 K). We simulate the period from 2005 to 2016. The spin-up time is about 2 years, similar to Schoeberl and Dessler (2011). In pure LTF simulations, the equilibrium number of tracked air parcels is about 400,000 (500,000 in Schoeberl and Dessler, 2011) but in the case of experiments including small-scale mixing parameterizations, that number increases to  1.6 million (Table 1) due to the spawning of new parcels inside the domain which adds up to the release at $\theta = 360$ K.

### 2.3 Experiments

We performed five LTF experiments with CLaMS. A summary of all experiments is provided in Table 1. This set of experiments is configured in such a way that the tested parameterizations are added cumulatively, increasing the complexity of the simulations and number of included processes step by step.

#### 2.3.1 Pure trajectory (LTF) experiments: TRAJ, CHEM and CIRRUS

The first set of 3 experiments (called TRAJ, CHEM and CIRRUS) use a pure LTF with advective trajectories launched exclusively at $\theta = 360$ K. While they are based on the same trajectories and hence transport, they differ in their treatment of chemical and microphysical processes impacting water vapor.

In TRAJ, chemistry and detailed microphysics are essentially ignored. Air parcels are initialized with 50 ppmv water vapour at the launch level. Thereafter, water vapour in excess of saturation (100% relative humidity RH) is removed at each time step. From a microphysical point of view, this is equivalent to assuming instantaneous formation and fall-out of all ice particles at 100% RH. Note that this approach is, in practice, equivalent to setting to the Lowest saturation Mixing Ratio (LMR) encountered by the air parcel along its trajectory, as in Fueglistaler and Haynes (2005). Saturation mixing ratios over ice are estimated from the 6-hourly ERA-interim temperature and pressure following Murphy and Koop (2005).

In CHEM, the moistening effect of methane oxidation is included by applying the CLaMS chemistry module (Pommrich et al., 2014). The corresponding reactions are a significant source of water vapour in the middle and upper stratosphere. The methane mixing ratio at launch level is taken to be 1.7 ppmv, following Schoeberl et al. (2013). As in TRAJ, water vapour in excess of 100% relative humidity is removed at each time step. The CLaMS dehydration scheme (for details see Von Hobe et al., 2011) is configured equivalently to the LMR calculation in the basic TRAJ case. Therefore, an air parcel is set to saturation whenever its water vapour content is above 100 % of relative humidity (RH), following Marti and Mauersberger (1993), which is similar to Murphy and Koop (2005).

The third experiment, CIRRUS, applies the same initialization and simplified chemistry as CHEM. However, it also takes into account the ice phase (although simplified): in case of supersaturation, excess water vapour is instantaneously transferred to the ice phase, instead of being removed, as described in Von Hobe et al. (2011). Then, a mean (spherical) ice particle size and the corresponding settling velocity are computed using an empirically defined ice particle density (not temperature-dependent) based on in situ observations (Krämer et al., 2009). The calculated sedimentation length of the ice particles during one time step is compared to a characteristic length ($\sim$300 m, optimized by Ploeger et al. (2013)) to yield the fraction of ice removed from the air parcel (e.g., if the sedimentation length is a third of the characteristic length, 30 % of the ice is assumed to fall-out). If during the following time steps an air parcel turns out to be subsaturated and ice exists, all ice evaporates till the air parcel reaches saturation.

It should be noted that this simplified microphysics scheme does not resolve the nucleation and growth of the ice particles, only their sedimentation. Consistently, it also does not include the effect of temperature fluctuations due to gravity waves (Jensen and Pfister, 2004) unresolved in the reanalysis. Although those are ubiquitous (e.g. Podglajen et al., 2016; Schoeberl

et al., 2017), it is not straightforward to include them in the simplified microphysics scheme, in particular due to their complicated interaction with ice nucleation and ice crystal number density (Dinh et al., 2016; Jensen et al., 2016). Furthermore, earlier studies have argued that their impact on water vapour itself is marginal (Fueglistaler and Baker, 2006) and that they mainly affect the ice cloud cover (e.g., Schoeberl et al., 2016, 2018). We therefore refrain from including gravity wave induced temperature fluctuations, but regard them as an additional uncertainty for our study.

### 2.3.2 Experiments including small-scale mixing effects: SSMIX, VMIX and STANDARD

One of the key features of the CLaMS model is its parameterization of small-scale mixing processes (McKenna et al., 2002b; Konopka et al., 2004, 2007). This parameterization has been proposed to include the effects of small-scale mixing on tracer distributions, mainly in regions where large-scale flow deformations occur (Konopka et al., 2004; Pan et al., 2006; Konopka et al., 2007).

Details about the mixing parameterization and its consequence in terms of diffusivity can be found in McKenna et al. (2002b), Konopka et al. (2004), Konopka et al. (2007) and Poshyvailo et al. (2018), and we only briefly summarize the governing principle here. The relative position of each parcel and its nearest neighbour are tracked during advection by the reanalysis wind over a 24 hour time step. Due to vertical wind shear and horizontal deformation, the horizontal distance between the air parcel and its nearest neighbours after advection changes. If this distance falls below a threshold distance $r_- = r_0 e^{-\lambda_c \Delta t}$, both air parcels are merged into one parcel at the mid point. If the distance exceeds $r_+ = r_0 e^{+\lambda_c \Delta t}$, a new air parcel is inserted in the middle. This adaptive regridding is the core piece of the mixing scheme and ensures that the horizontal distance between parcels remains of the order of $r_0$ while allowing for some deformation. The composition of the new air parcel, i.e its mixing ratios of water vapour, methane and ice, is set to the average of the mixing ratios of the parcels that experienced mixing.

The fourth experiment, SSMIX, adds the small-scale mixing parameterization of CLaMS to the processes represented in CIRRUS. After a mixing event, the same dehydration scheme as in CIRRUS is applied again to remove the supersaturation that may have been introduced in new air parcels due to the mixing and by-passing cold temperatures. Transient temperature fluctuations in turbulent layers are neglected, but are likely short-lived and not causing a significant effect.

The fifth experiment, called VMIX, includes enhanced tropospheric mixing recently developed by Konopka et al. (2019). This additional parameterization relates tropospheric mixing to unresolved convective instability. In this VMIX experiment, air parcels with a (moist) Brunt–Väisälä frequency, $N_m^2$, larger than a predefined value, $N_c^2 = 0.0001 s^{-2}$, undergo tropospheric mixing with their nearest neighbours: their chemical composition is changed to the averaged mixing ratios of all the parcels involved in the mixing process. Contrary to the standard mixing scheme, this procedure does not change the position of the air parcels.

Finally, besides the above experiments based on the LTF set-up, we consider the full-blown chemistry transport model standard version of CLaMS (STANDARD) (Diallo et al., 2018; Tao et al., 2019). This simulation includes the same parameterizations as SSMIX, but has a different initialization. The air parcels are released at the beginning of the simulation throughout the domain, covering both the troposphere and stratosphere, with a horizontal resolution of about 100 km in the UTLS (for details see e.g., Konopka et al., 2019). Once released, trajectories of air parcels are computed using reanalysis horizontal wind

fields and diabatic heating rates for vertical transport. When air parcels are in the troposphere below about 500 hPa, their water vapour content is interpolated from ERA-interim while methane is derived from ground-level observations. The CLaMS dehydration and chemistry schemes are applied, configured consistently with the CIRRUS experiment. Note that contrary to the LTF technique, the boundary of the model is the surface such that air parcels are not filtered out when they reach below 250 hPa or above 1800 K (as in the LTF experiments), and the water vapour content of air parcels at 360 K is not fixed uniformly

to 50 ppmv but calculated consistently in the model. Here we will refer to this set up as "Stratosphere-Troposphere Filling" (ST-Filling). Further details of the initialization can be found in Pommrich et al. (2014).

## 2.4 Aura MLS observations

Satellite observations of water vapour mixing ratios in the LS from Aura Microwave Limb Sounder (Waters et al., 2006) are used to further assess the results. We use the version 4.2 of the water vapour data from MLS (Lambert et al., 2015) which has

been fully described in (Livesey et al., 2018). These water vapour products have been validated in several studies and recently have been part of a climatological overview of the Asian Monsoon anticyclone (Santee et al., 2017). Here, Aura MLS data and CLaMS data have been compiled on the same regular latitude-longitude grid as the one used by the experiments. In particular, we use the MLS data on 100 hPa and 82 hPa pressure levels and compare to the simulated water vapour distributions at 100 hPa and 80 hPa, respectively. Since we are interested in differences between set-ups, we did not apply the averaging kernels to the

model outputs to avoid potential smearing out of fine scale patterns. We note that applying the averaging kernels of MLS does not change the pattern of water vapour in the lower stratosphere during boreal summer.

## 3 Results

### 3.1 Boreal summer climatology of lower stratospheric water vapour

Figure 1 (left column) shows the climatological water vapour distribution at 100 hPa during June–August (JJA) over the 2007-

2016 period in the different CLaMS experiments and MLS observations (i.e. excluding the spin-up). A similar figure for the 80 hPa is presented in Fig. 2.

All experiments (Fig. 1b–f) reproduce the main characteristics of the water vapour distribution found in MLS. The contrast between the dry tropics and subtropics and the moister mid and high latitudes seen in MLS is present in all simulations. Furthermore, all experiments, including TRAJ, exhibit a local maximum in the Asian Monsoon Anticyclone (AMA). This

consistency between the different experiments emphasizes the key role of transport through the large–scale temperature field in causing this feature (as found in e.g. James et al., 2008). However, there are also important differences between the experiments and with MLS observations. Compared with MLS (Fig. 1a), experiments with the LTF scheme (Fig. 1b–f) underestimate the water vapour content in the moistest regions. This underestimation reaches 1.5 to 2 ppmv in the case of TRAJ, pointing either to biases in large–scale transport and temperatures in ERA-interim or to missing processes, as expected. The dry bias is reduced

in the more sophisticated experiments which include more processes. In addition, there is a misrepresentation of the North

American Monsoon Anticyclone (NAMA) in all experiments, which tend to show a weaker maximum shifted towards the eastern and central Pacific with respect to observations. It should be noted that at 80 hPa the agreement between STANDARD and MLS is much better (Fig. 2). Larger differences at 100 hPa are likely related to the fact that this lower pressure level is partly in the stratosphere and partly in the troposphere (e.g., in the Asian monsoon where the tropopause is frequently above 100 hPa).

Small biases in reanalysis tropopause height can therefore cause large biases in simulated water vapour at 100 hPa, while at 80 hPa their effect is marginal. Nevertheless, we focus our results at 100 hPa here as this is the most frequently considered level for investigating the monsoon UTLS, and further our goal is not to identify a best-case simulation scenario but to estimate the effects of different processes from the sensitivity simulations, and these estimates are very similar on 100 and 80 hPa.

In order to separate the effect of each parameterization, Figure 1 (right column) displays the differences between pairs of
experiments which share the same configuration except for one single process parameterization. Thus, Fig. 1h (CHEM minus TRAJ) isolates the impact of methane oxidation, which is only included in CHEM (see Table 1). Similarly, Figure 1i, j and k show the impact of the simplified ice microphysics (cirrus) parameterization (CIRRUS-CHEM), small-scale mixing (SSMIX-CIRRUS) and enhanced tropospheric mixing (VMIX-SSMIX), respectively.

Regarding methane oxidation, Figure 1h shows a water vapour increase of around +0.1 ppmv over the tropics and subtropics
that reaches +0.2 ppmv over high latitudes, as expected. As methane oxidation occurs at mid-stratospheric levels (Randel et al., 1998), its impact on the water vapour distribution at 100 hPa (Fig. 1h) is a consequence of air parcels moving downward from those altitudes following the downwelling branch of the Brewer–Dobson circulation at high latitudes. During boreal summer, the downward circulation is stronger in the southern hemisphere, which explains the larger increase of water vapour in this region. Once air parcels reach the lower stratosphere at high latitudes, some of them may reach the troposphere below 250
hPa, where they are removed from the simulation, while others follow the residual meridional circulation giving rise to the observed subtropical and tropical enhancements of water vapour at 100 hPa. This weak meridional transport was also observed by Ploeger et al. (2013) and Poshyvailo et al. (2018). A similar impact of methane oxidation can be found at 80 hPa in spite of the stronger meridional gradient at this pressure level (Fig. 2).

Including our simplified representation of ice microphysics (Fig. 1i ) results in a further moistening of the LS ranging from
+0.4 ppmv to +0.6 ppmv over most regions. These values are in agreement with the global increase of +0.5 ppmv found by Ploeger et al. (2013) and exceed the effect related to methane. Moreover, Figure 1i shows that the effects of ice are especially large in the AMA (about +0.8 ppmv at 100 hPa), enhancing the moisture anomaly in the monsoon UTLS. This signature is also found at 80 hPa but with slightly weaker values compared to 100 hPa (Fig. 2i).

Small-scale mixing has a similar impact (Fig. 1j), increasing water vapour at latitudes north of 30°S and especially in the
AMA. Outside of the AMA, the water vapour increase linked to small-scale mixing is slightly weaker than that attributed to ice microphysics (+0.1 to +0.4 ppmv). The local impact on water vapour in the AMA region, however, is stronger and reaches values above +0.9 ppmv at 100 hPa. This strong moistening effect can be attributed to the fact that mixing processes are more frequent in regions with large–scale flow deformations, which are mainly located in the surroundings of the subtropical jet (Konopka and Pan (2012), Poshyvailo et al. (2018)) and hence in the AMA.

The effects of the enhanced mixing in the troposphere as represented in the VMIX experiment are shown in Fig. 1k. An increase in water vapour of up to +0.1 ppmv occurs almost everywhere north of 30°S, with again a relatively stronger impact in the AMA, with differences larger than +0.3 ppmv. This stronger influence in the AMA compared to other regions is also found at 80 hPa but weaker than at 100 hPa. Compared to other processes, VMIX shows a weak increase of water vapour in the AMA related to the enhanced tropospheric mixing.

The reason behind this increase of lower stratospheric humidity with the mixing parameterizations in SSMIX and VMIX ultimately lies in the bypassing of cold traps which air parcels would have otherwise encountered along their slow ascent and the associated horizontal wandering (for further details see App. A).

The 80 hPa level shows a similar spatial distribution of water vapour differences to that at 100 hPa, with values peaking again in the AMA (Fig. 2j). However, the relative strength of this maximum (+0.5 ppmv) is weaker than at 100 hPa.

Since mixing changes the temperature encountered by air parcels, the impact of temperature-dependent processes, such as ice microphysics, is altered. To evaluate this effect, we have run a two-year simulation, hereafter called "VMIXnocirrus", in which all the water vapour in excess of saturation is instantaneously removed instead of being transferred to the ice phase. The difference between VMIXnocirrus and VMIX shows the impact of ice transport, as CIRRUS-CHEM, but with mixing being applied (see Fig. A2). Comparing Fig. A2 with Fig. 1i, the moistening effect caused by the inclusion of a simple ice

microphysics scheme is amplified with mixing. Nevertheless, the spatial pattern resembles that of the experiments without mixing and peaks in the AMA region (with +1 ppmv). We interpret this enhancement of the moistening as caused by the vertical transport of both ice and a larger vapour content by mixing. By effectively bypassing cold traps, mixing favors ice sublimation, which (1) directly increases the water content and (2) decreases the size of the remaining ice particles and hence their settling velocity, thereby increasing their residence time and the possibility of subsequent sublimation in warmer regions.

Together with the transport of the larger vapour background content associated with this set-up, this leads to an enhanced moistening.

Finally, the sensitivity of the LS water vapour to the boundary condition imposed in the LTF set-up is assessed by comparing SSMIX to the full-blown CLaMS STANDARD experiment. As a reminder, STANDARD uses the same parameterizations as SSMIX, but calculates transport throughout the troposphere using ERA-Interim water vapour values as the lower boundary

condition below about 500 hPa. Thus, contrary to the LTF initialization, the water vapour content of the air parcels at 360 K depends on the transport properties of the air parcels reaching that level. Figure 1 depicts the water vapour distribution obtained for the STANDARD simulation (panel g) and its differences with respect to SSMIX (panel l). The STANDARD simulation exhibits a much wetter stratosphere than SSMIX, which leads to a weak overestimation of the water vapour compared to MLS, in particular in the AMA region. However, at 80 hPa there is good agreement between the water vapour field simulated in

STANDARD and MLS (Fig. 2g). Figure 1l shows that the main differences caused by the LTF scheme are not centered on the AMA region but on both the Western and Eastern parts of the North Pacific and in the 20°–30°S latitude band. At 80 hPa, differences occur in the same latitude band and expand zonally. This implies that the global effect of the LTF set-up is to dry the stratosphere compared to the STANDARD simulation, in particular at the edges of the tropics, and with smaller differences in the AMA.

Concerning the water vapour maximum found over the NAMA in MLS observations, we found that its spatial pattern is not well reproduced in any of the experiments (Fig. 1). The maximum is shifted to the West compared to MLS over the Eastern Pacific and, except for the STANDARD simulation, which shows water vapour values in the NAMA close to the observations, all other experiments display much lower values. The mixing parameterization has a much weaker effect in the NAMA compared to the AMA, which suggests that the weaker anticyclonic monsoon circulation over that region produces a

weaker deformation of the main flow leading to less mixing between air masses. The NAMA water vapour maximum seen in MLS is known to be more challenging to simulate than the AMA (e.g., Ueyama et al., 2018).

## 3.2    Sensitivity to assumptions in the microphysics

Ice microphysics in the UTLS is a complex issue, which requires sophisticated models (Jensen and Pfister, 2004; Ueyama et al., 2018) as well as a series of assumptions regarding the nature of ice nuclei, the shape of ice particles and their dynamical

environment including convective detrainment and gravity waves. We have here considered a simple representation of the microphysics (in CIRRUS and related model experiments), in which the ice phase and water vapour are kept in thermodynamic equilibrium and ice particles sediment. However, both laboratory experiments and in situ observations (e.g. Krämer et al., 2009; Krämer et al., 2020) show the common occurrence of large supersaturations under clear sky conditions, which is related to a delay of ice nucleation to high supersaturations at low temperatures.

For a simple test of the sensitivity of our results to a potential supersaturation threshold required for ice formation, we performed a second CIRRUS (pure LTF) simulation, in which ice formation is delayed to a relative humidity of 150 %. If this value is reached, all vapour in excess of saturation is condensed into the ice phase (so that the parcel is at 100 % relative humidity). Figure 3 compares the distribution of water vapour during boreal summer averaged for the period 2005-2010 for CIRRUS with ice formation threshold 100% and 150%. Our results show that by allowing further transport of water vapour

before ice formation, CIRRUS150% results in a more humid LS everywhere. This moistening effect of increasing the saturation level is proportional to the local saturation mixing ratio and especially large in regions in which ice microphysics has a strong signature, such as the AMA and NAMA.

It should be kept in mind that in situ ice formation, as represented here, is only one of the many processes through which ice influences the water vapour content. Ice may be detrained in the UTLS from deep overshooting convection and evaporate

afterwards, resulting in a net hydration of the UTLS (Corti et al., 2008). However, Jensen et al. (2020) concluded from observations that this effect is small, except in the NAMA, which may partly explain the poor model results there. On the other hand, convection also directly influences the vapour phase, an issue discussed in Sect. 4.1.

## 3.3    Variability of water vapour over monsoon regions

### 3.3.1    Seasonal variability

Figure 4 shows the annual cycle at 80 hPa (top) and 100 hPa (bottom) of the simulated and observed water vapour in the AMA (left column) and NAMA (right column), averaged over the period 2007–2016. For a proper comparison of the annual cycle

the peak-to-peak change between the different experiments, we have subtracted the respective April average (hereafter referred to as the offset) from each time series. April was chosen because this is when, in most experiments, the water vapour content is closest to its minimum in the AMA and NAMA.

Figure 4 shows that over both the AMA and NAMA regions and at both 100 hPa and 80 hPa, all experiments represent the observed increase of water vapour during boreal summer. The peak water vapour shows a 1–2 months delay at 80 hPa with respect to 100 hPa. Figure 4b reveals that CIRRUS, TRAJ and CHEM better represent the amplitude of the seasonal cycle of water vapour at 100 hPa in the AMA, according to MLS observations, although they underestimate the absolute value in summer (see their average for April and also Fig. 1). At 80 hPa, however, the simulations including small-scale mixing are in better agreement with MLS.

The CIRRUS experiment slightly overestimates the peak-to-peak amplitude at 100 hPa, which is about 0.3 ppmv higher than in MLS observations (Fig. 4b). CIRRUS also shows a higher offset than TRAJ and CHEM, related to a higher annual cycle minimum. Consequently, CIRRUS shows a water vapour distribution that is closer to the observations not only during the monsoon season, as shown in Fig. 1, but throughout the year, compared to TRAJ and CHEM.

Figure 4b shows a very steep water vapour increase in the AMA between June and August for SSMIX and VMIX, resulting in an overestimation of the amplitude of the annual cycle at 100 hPa (+0.8 ppmv compared to MLS and +0.9 ppmv compared to TRAJ). Furthermore, at this pressure level the differences between VMIX and SSMIX are slightly larger during the monsoon season, which is linked to the enhanced tropospheric mixing in VMIX and might be related to the impact of enhanced convective updrafts over the monsoon region during summer. Also, Figure 4b shows that the water vapour decrease, observed from September onward, is also faster in SSMIX and VMIX than in MLS observations. Thus, the good agreement between SSMIX and VMIX in the AMA during the monsoon season (Fig. 1e and f) can be attributed, on the one hand, to an increase in the minimum value over the annual cycle (as is evident from the April averages in Fig. 4b) which nevertheless remains underestimated, and on the other hand to an overestimation of the water vapour increase during the monsoon July and August.

At 80 hPa, on the other hand, the annual cycle is better reproduced by SSMIX and VMIX than by the pure LTF models (Fig. 4a). Thus, while SSMIX and VMIX show a water vapour increase and a peak amplitude very close to the observations, the "no-mixing" experiments clearly underestimate the amplitude of the annual cycle and typically exhibit a slower water vapour increase in summer and a slight delay (1-2 weeks) of the annual maximum.

Finally, since the STANDARD experiment includes the same process parameterizations as SSMIX, it is not surprising that it also shows an overestimation of the water vapour increase in the AMA at 100 hPa, with a peak amplitude that is about 0.5 ppmv larger than in MLS (Fig. 4b). As for SSMIX and VMIX, STANDARD also results in a faster water vapour decrease from September onward at this level. However, despite a very similar behaviour to SSMIX/VMIX at 80hPa, STANDARD exhibits a weaker water vapour increase than SSMIX at 100 hPa.

Figures 4b evidences that the STANDARD depicts the largest offset respect to MLS. Nevertheless, the difference in water vapour between STANDARD and MLS increases at 100 hPa during the mature phase of the AMA. This can be most likely attributed to the excessive amount of water vapour created by the small-scale mixing at the beginning of the monsoon season, as is the case for SSMIX and VMIX as well.

In the NAMA region, the annual cycle in water vapour is more consistent between the different experiments as compared to the AMA, but differences to MLS are larger (Fig. 4c and d). The increase of simulated water vapour occurs from May to September and is weaker than in MLS observations. This leads to a peak-to-peak amplitude underestimated by 0.2 ppmv to 0.4 ppmv depending on the experiment, and to a delay of about 1 month to 6 weeks in the annual cycle maximum. Also, the impacts of the non-instantaneous removal of ice and of small-scale mixing in the NAMA region are quite uniform throughout the year and do not exhibit intensification during the monsoon season. This suggests a minor impact of small-scale mixing processes in the NAMA compared to the AMA. Although both SSMIX and VMIX show a wetter NAMA than CIRRUS (Fig. 1), according to Fig. 4d this is due to an uniform impact of small-scale mixing processes throughout the year rather than a peak during the monsoon season. As previously mentioned, small-scale mixing depends on deformations of the large–scale flow. The large–scale circulation of the NAMA is less confined than the AMA (Gettelman et al., 2004), which could render this region more sensitive to overshooting convection (not included in our experiments) in comparison with the AMA. Indeed, also the larger and consistent differences of all model experiments to MLS point to a significant role of convection, the commons process neglected in all simulations, for moistening the NAMA. Furthermore, the STANDARD experiment shows a similar behaviour during the monsoon season, evidencing that the differences in the initialization scheme have a very limited influence on the annual cycle of the water vapour over the NAMA.

The annual cycle at 80 hPa (Fig. 4c) shows lower water vapour increases, peak-to-peak amplitudes and a delayed maximum for all experiments compared to the observations. At this level, the experiments which best match the observed annual cycle are STANDARD, SSMIX, VMIX and CIRRUS, i.e. those exhibiting a higher water vapour content at 100 hPa (Fig. 1). Note that even for those experiments significant differences to MLS remain (about 0.5 ppmv).

### 3.3.2 Subseasonal variability

In order to assess the representation of water vapour variability beyond the seasonal cycle in the AMA, Figure 5a–f depicts deseasonalized daily anomalies of water vapour during 2007–2016 for each experiment and MLS observations together with the respective correlations. In order to evaluate the experiments for the entire monsoon season (from May to September, MJJAS) and the mature phase of the monsoon season (June-July-August, JJA), these correlations are computed for both periods. All LTF experiments exhibit statistically significant correlations and the correlations tend to increase with the complexity of the experiment (i.e. the number of processes included). Thus, the simpler configurations TRAJ and CHEM have the lowest correlations of 0.51 (JJA) and 0.62 (MJJAS) (p<0.025). Note that these values are consistent with those obtained by Zhang et al. (2016) in similar experiments using ERA-interim (their Fig. 6). Although lowest among all experiments presented here, the still significant correlations between TRAJ (CHEM) and MLS support the idea that large–scale cold point temperature variability is the main factor controlling water vapour variability in the Asian monsoon UTLS, as also argued by Randel et al. (2015); Zhang et al. (2016). Furthermore, the small difference between correlations in TRAJ and CHEM experiments reveals that methane oxidation is irrelevant to water vapour variability in the AMA.

Including the simple parameterization of in situ ice formation and evaporation (CIRRUS) slightly improves the correlation during both JJA and MJJAS. This improvement is even higher when small-scale mixing processes are also included. Thus,

among all LTF experiments the highest correlations are obtained for SSMIX (r=0.64/0.69 for JJA/MJJAS , p<0.025) and VMIX (r=0.62/0.66 for JJA/MJJAS , p<0.025). This result manifests the importance of mixing for the simulation of a realistic water vapour variability in the AMA, despite the overestimation of the water vapour increase at the beginning of the monsoon season at 100 hPa found in Fig. 4a and b. Comparing SSMIX with VMIX shows that the enhanced tropospheric mixing, which has only a mild impact on the water vapour distribution (Fig. 1f and k), does not improve the simulation of water vapour variability. Overall, the experiments including mixing do a significantly better job in simulating the sub-seasonal variability (correlations of about 0.6) than the pure LTF experiments (TRAJ, CHEM and CIRRUS, correlations with MLS slightly above 0.5). Hence, including mixing processes improves the simulation of water vapour variability in the AMA on sub-seasonal timescales. Finally, Figure 5f shows evidence that, for both periods, the STANDARD simulation correlates best with MLS, reaching values of 0.76 and 0.74 (p<0.025) for JJA and MJJAS, and significantly improves the intra-seasonal variability in the AMA.

In the NAMA, simulated deseasonalized daily anomalies of water vapour correlate relatively well with MLS for both periods, reaching even higher correlations than in the AMA (Fig. 5g–l), despite the poor representation of the water vapour climatology in this region. The STANDARD simulation shows the highest correlation with MLS (r=0.83, p<0.025 in JJA) followed by the LTF experiments that include small-scale mixing (SSMIX, 0.77 and VMIX 0.78 in JJA). The lowest correlation is achieved by TRAJ (r=0.73 in JJA), which is yet very similar to the highest correlation achieved in the AMA region. Again, the TRAJ experiment already shows high correlation with MLS, indicating that temperature is the main control factor for intra-seasonal variability also in the NAMA. The reason why correlations in the NAMA are higher than in the AMA is likely related to the fact that processes other than large–scale temperature variability play a larger role when the anticyclonic circulation and related confinement are strong enough, as in the AMA.

## 4  Discussion

### 4.1  Convective moistening

Another important process for the UTLS water vapour budget is ice and moisture transport by convection (Dessler and Sherwood, 2004; Dessler et al., 2016). Although some of the experiments presented here include processes whose particular parameterizations may contribute to convective moistening (e.g., small-scale mixing), a direct simulation of convection is not present in these simulations.

To further investigate the additional effects of convection on the monsoon water vapour budget, we performed a modified TRAJ experiment, hereafter called CONV, in which this process is taken into account, following an approach similar to that of Ueyama et al. (2018). In CONV we use the trajectories of TRAJ to compute the LMR of the air parcels. At every timestep it is checked whether an air parcels is located inside a cloud, i.e. at a pressure level below that of the cloud top. If this is the case, the air parcel's water vapour is set to the saturation mixing ratio (100 % relative humidity), according to the temperature that the ERA-Interim reanalysis attributes to the location of the air parcel. This process corresponds to hydration if the air parcel is initially subsaturated and dehydration if it is supersaturated (see Ueyama et al., 2018). We use ISCCP B1 (GridSat-B1) Infrared

Channel Brightness Temperature combined with ERA–interim data to determine cloud top heights (Knapp, 2014), following the methodology of Tissier and Legras (2016). Their approach is similar to the one employed by Ueyama et al. (2018) and assumes that the temperature at cloud top is equal to the temperature of the environment estimated from the reanalysis. The resulting altitude is shifted upward by 1 km to correct for biases in infrared cloud top temperature (Minnis et al., 2008). There are a few differences, however, between the two methods: first, contrary to Ueyama et al. (2018), Tissier and Legras (2016) do not distinguish convective cores from in situ-formed cirrus clouds and include both for cloud top determination. The impact of this difference should be small as long as the cirrus clouds are sufficiently thin. Second, in the case of brightness temperatures lower than the local tropopause temperature, we assume that convective parcels rise adiabatically from 40 hPa below the tropopause, whereas Ueyama et al. (2018) take a mixture of tropopause (70%) and environmental air (30 %). Therefore, our estimated cloud top altitudes may be low biased compared to theirs. Finally, note that the ISCCP B1 has slightly lower horizontal resolution (∼8 km vs 4 km) than the dataset of Ueyama et al. (2018) but similar temporal resolution (3-hourly). Figure 6 shows the water vapour distributions at 100 hPa of TRAJ (Fig. 6a), CONV (Fig. 6b) and the differences between both of them. The two experiments result in a very similar water vapour distribution, with a slight moistening effect caused by convection, mainly at mid and high latitudes. These results indicate that even when a convective event occurs, the water vapour is set by temperatures experienced by the air parcels after convection. This result is in agreement with Randel and Park (2006); Randel et al. (2015), but not in line with Ueyama et al. (2018). However, it should be kept in mind that Ueyama et al. (2018) focused on the analysis of a single 7-day convective event during summer 2007, whereas we consider the entire summer (June–August) for 2005-2009. Thus, while their main conclusion is that infrequent deep convection reaching above 380K causes a strong moistening of the LS, our results show that the climatological impact of these events is likely very weak.

We have also considered only the 7-days Ueyama et al. (2018) to check if we were available to reproduce the same results with TRAJ and CONV as their simulations without and with convection, respectively. However, in our case TRAJ produces a maximum of water vapour in the AMA that is not observed in their non convective experiment. This suggests that there are additional features, further than the differences mentioned before, that make the comparison between the experiments in Ueyama et al. (2018) and ours difficult. Another limitation of our approach in CONV is that we have not taken into account the role of convective ice. According to Wang et al. (2019), the main impact of convection on the LS water vapour, occurs through the injection of ice. The latter assertion, however, is contrary to the conclusions of Ueyama et al. (2018). These differences between the different studies highlight the a large existing uncertainty about the role of convection for LS water vapour. A deeper analysis of this issue should be considered in future studies.

## 4.2 Sensitivity of water vapour to the LTF set-up

In the LTF experiments previously described we have used the same longitude-latitude grid to release new air parcels at the same initial potential temperature of 360 K. This configuration, which has been chosen following the procedure of Schoeberl and Dessler (2011), might have an effect on our results. In fact, Schoeberl et al. (2013); Wang et al. (2019) use 370 K as the initial potential temperature level for air parcels released in the Asian Monsoon region. They argue that because the LZRH is higher over the AMA than in other regions, many air parcels released at 360 K may descend and are removed from the

simulation. Figure 7a shows the normalized distribution of air parcels in TRAJ at 100 hPa centered in the AMA for JJA in 2007. As we could expect from Schoeberl et al. (2013); Wang et al. (2019), the AMA is characterized by a lower density of air parcels. Because a low number of air parcels could condition the robustness of the results, this raises the question of the possible impact of the number of air parcels in the AMA on the water vapour distribution in this region. To test this potential sensitivity to the number of air parcels, we have performed two additional TRAJ experiments in which either the number of air parcels newly released has been increased (TRAJ-denser) or the initial potential temperature (TRAJ-370 K) has been changed. In TRAJ-denser the number of air parcels is increased by releasing them on a higher resolution grid (2.5°-longitude x 1°-latitude). By contrast, in TRAJ-370 K the initialization grid maintains the same resolution as in TRAJ but air parcels are launched at the $\theta = 370$ K surface, instead of at 360 K.

Figure 7(b,c) shows the relative number of air parcels in the AMA at 100 hPa of TRAJ-denser and TRAJ-370 K with respect to TRAJ. In addition, Figure 7(d-f) shows the water vapour distributions of TRAJ, TRAJ-denser and TRAJ-370 K for the same season and year. In both experiments, the density of air parcels has increased notably with respect to TRAJ. There are four times more air parcels in TRAJ-denser (Fig.7b) and twice in TRAJ-370 K (Fig.7c) than in TRAJ. As in TRAJ-denser more air parcels are released inside the AMA, a larger number of them reach the 100 hPa level. However, this larger number of air parcels is not accompanied by an enhancement of water vapour (Fig.7e). Therefore, we conclude that increasing the resolution of the initialization grid used in our LTF experiments does not have an impact on our results, which means that the resolution and particle number of the TRAJ experiment are high enough to adequately capture the spatial variability of the temperature field and its impact on water vapor. By contrast, TRAJ-370 K exhibits a moister water vapour distribution in the AMA than the original TRAJ experiment (Fig.7f). This might be explained by the fact that the AMA shows a stronger anticyclonic circulation at 370 K than at 360 K (Bergman et al., 2013), giving rise to the uplift of more air parcels through the inner core of the AMA. The stronger confinement of this region can allow a greater number of air parcels to avoid the coldest regions, which are located at the southeastern flank of the AMA. Thus, when the air parcels spawn to other regions at 100 hPa level due to the weakening of the anticyclone circulation, they transport a larger water vapour content than air parcels reaching the same pressure level in TRAJ. This widening of the vertical transport conduit of air parcels when being uplifted in the Asian monsoon anticyclone is consistent with the main transport pathway proposed by Bergman et al. (2013).

These results evidence that our LTF experiments have a sufficient number of air parcels to be statistically significant. Furthermore, they point out that the selection of a higher initial potential temperature in the AMA has an impact on water vapour distributions, likely related to the larger exposure of the chosen potential temperature level to stronger confinement in the AMA. Thus, we conclude that the selection of the initial potential temperature should not only take into account the level of zero radiative heating, but also the strength and confinement of the AMA.

These sensitivity tests of the domain filling technique to a different selection of arbitrary chosen parameters do not cover possible impacts of the LTF scheme itself in our water vapour results. To study this we compare a LTF experiment, SSMIX, with a non LTF experiment, STANDARD. Despite the similar set of parameterizations included, the STANDARD and SSMIX experiments significantly differ and STANDARD agrees better with MLS regarding seasonal and intraseasonal variability in

the AMA and NAMA. The remaining discrepancies might be due to: i) the initial water vapour content of the air parcels and ii) the filtering of air parcels below 250 hPa and above 1800 K, in particular the absence of a troposphere.

To test point i), we performed an additional experiment configured as SSMIX, but doubling the initial water vapour content of air parcels from 50 to 100 ppmv. Figure A3 in the appendix shows that the distribution at 100 hPa is not affected by this change of the initial condition. Note however that this does not suggest that the water vapour is entirely insensitive to the lower boundary condition, as explained below.

In order to further investigate this result, Figure 8 depicts the 360 K water vapour maps (corresponding to the level of initialization of SSMIX) for MLS observations, STANDARD and the two SSMIX experiments. Note that the water vapour mixing ratio in the SSMIX is significantly lower than the initialization value. This is due to the presence of older air parcels which have been released at earlier time steps, have undergone dehydration and been transported into the 360 K layer from above. This stagnation and return of old air parcels is allowed as the filtering occurs at 250 hPa, which is below the initialization level. Therefore, we conclude that the initialization value in LTF experiments cannot be considered as a lower boundary condition for LTF simulations in the Asian Monsoon.

It turns out that the water vapour variability at 100 hPa is, indeed, sensitive to this lower boundary condition. This is indicated by the high correlations between the water vapour at 360 K in the AMA region and local $H_2O$ at 100 hPa (Fig. 8 e–h), which peak over the Asian Monsoon. The significantly different lower boundary conditions shown in (Fig. 8 a–d) thus likely cause parts of the observed differences between STANDARD and SSMIX at 100 hPa. In the standard LTF approach, this boundary condition is not directly set because of the mixture of old and young air parcels making up the air masses at 360 K.

While this issue may be interpreted as a too dry lower boundary condition, it is in the end related to missing transport pathways in the lower part of the simulated domain. This interpretation is supported by Fig. 9a which shows the relative percentage of air parcels simulated in STANDARD with respect to SSMIX. STANDARD shows a higher number of air parcels than SSMIX throughout the domain, and in particular in the AMA and the SH subtropics (Fig. 9a). Not only does STANDARD simulate more air parcels in the AMA, these parcels are also wetter, as shown in the probability density distribution (PDF) of the water vapour content of the air parcels (Fig. 9b). These differences suggest a lack of vertical transport of moist tropospheric air parcels from lower levels in the LTF initialization (where these air parcels are removed). Besides vertical transport, it is likely that inhibited horizontal entrainment also plays a role, since at the 360 K initialization level the inner anticyclone core is, to some degree, isolated from the surrounding areas, as shown in Garny and Randel (2016).

## 5   Conclusion

In this study, we compared numerical Lagrangian transport simulations based on the forward domain filling technique developed by Schoeberl and Dessler (2011) with the CLaMS model (McKenna et al., 2002b, a) in order to assess the impact of methane oxidation, ice microphysics, small-scale mixing and enhanced tropospheric mixing on the water vapour distribution in the lower stratosphere during boreal summer. A particular focus was laid on the Asian (AMA) and North American Monsoon Anticyclones (NAMA) in the UTLS.

In agreement with previous work (e.g., James et al., 2008; Schoeberl and Dessler, 2011), we find that simple last-dehydration-point LTF modeling based on large–scale reanalysis temperature and wind fields can qualitatively reproduce the water vapour signal in the AMA and its variability, but with simulated mixing ratios dry-biased. Furthermore, while our modeling set-up reproduces well the water vapour signal in the AMA, the location and amplitude of the NAMA maximum are less well reproduced.

While the effect of methane oxidation is small, a simplified representation of ice microphysics significantly moistens the LS. The magnitude of the water vapour enhancement largely depends on microphysical assumptions. A new finding of our study is that small-scale mixing processes, as parameterized in CLaMS depending on shear in the large-scale flow, has a strong impact on water vapour in the AMA region. A sensitivity simulation evaluating convective hydration suggests that its pattern is different from that of small-scale mixing and not particularly strong in the AMA, which tends to confirm the distinct signature of mixing. Interestingly, we find that the impact of changing microphysical assumptions also varies depending on the presence of mixing. This suggests that mixing is an important process to understand boreal summer water vapor. For a more complete picture of the UTLS boreal summer water vapour budget, future research should focus on investigating the impact of mixing on water vapour isotopes and high altitude cloud cover.

*Data availability.* Data used for experiments are available upon request from authors NP (npplamar@upo.es) and FP (fploeger@fz.julich.de). MLS H2O version 4.2 data can be obtained from the MLS website https://mls.jpl.nasa.gov (last access: 28 August 2020).

## Appendix A: Effect of mixing on bypassing cold traps

This hypothetical situation is illustrated in Fig. A1. Given two air parcels, "A" and "B", at different altitudes but close enough, they mix together into "C". In case C is supersaturated after mixing, the microphysics of ice turns it into saturation, forming ice particles with the excess water vapour. However, this saturation value would be higher than the corresponding to the minimum temperature in the vertical profile considered, Tmin. Therefore, the water vapour of C would not be set by Tmin, but by the temperature at its altitude. In case Tmin represents the temperature of the CPT, then water vapour has been transported to higher altitudes avoiding the CPT and giving rise to an increase of water vapour over most regions, but especially where the mixing is stronger.

*Author contributions.* NP, AP and FP designed the experiments. NP and AP performed the experiments with CLaMS. NP performed the data analysis. NP, CP, AP and FP contributed to the discussion of results. NP and CP wrote the text. CP, AP and FP made the final review.

*Competing interests.* The authors declare that they have no conflict of interest.

*Acknowledgements.* We would like to thank Bernard Legras for performing an experiment with TRACZILLA and share with us his work. We thank the Institute of Climate from the Research Center of Jülich and, in special, Martin Riese, for their scientific, technical and financial support. This research was funded by the Spanish Ministerio de Economía y Competitividad through the project Variabilidad del Vapor de Agua en la Baja Estratosfera (CGL2016-78562-P).

585

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

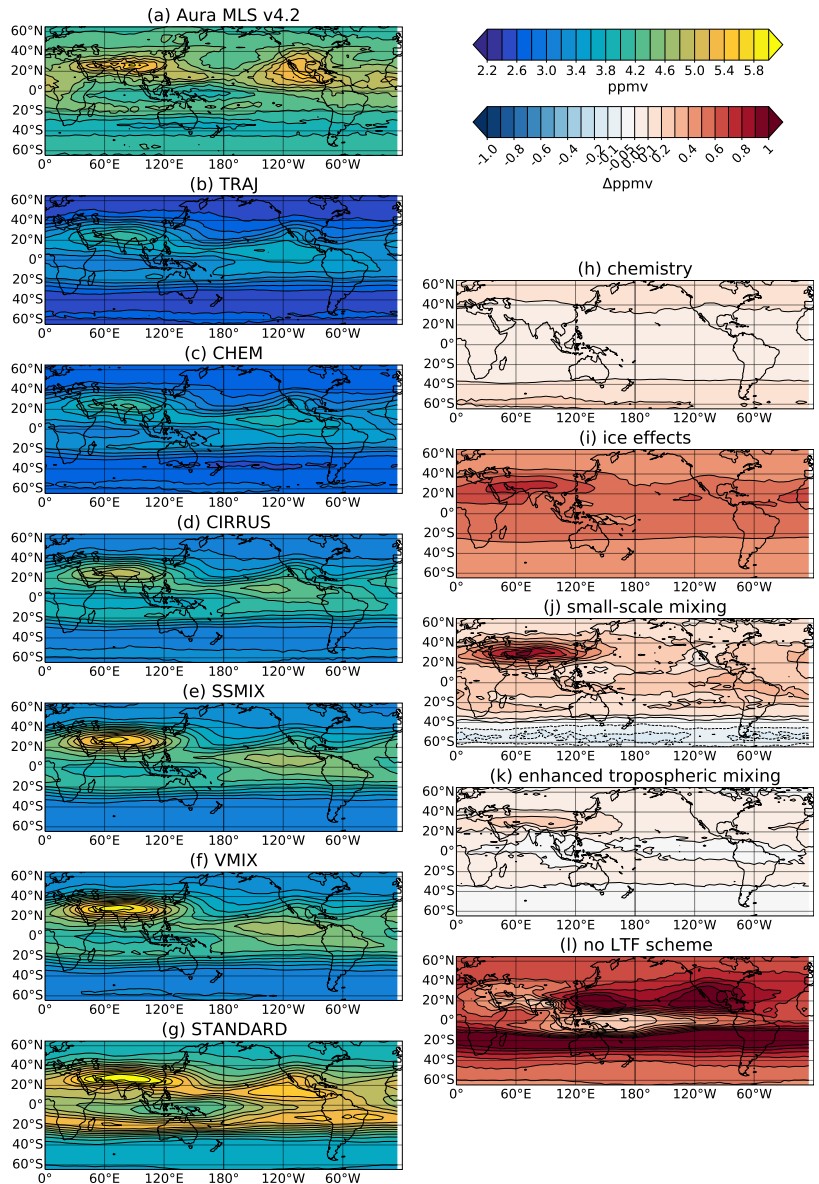

**Figure 1.** (Left column) Climatology of water vapour distribution at 100 hPa during boreal summer (June-July-August) for the period 2007–2016 from **(a)** Aura MLS v4.2 observations, **(b)** TRAJ, **(c)** CHEM, **(d)** CIRRUS, **(e)** VMIX, **(f)** SSMIX and **g)** STANDARD simulations. (Right column) Isolated effect of **h)** methane oxidation, **i)** cirrus, **j)** small-scale mixing, **k)** enhanced tropospheric mixing and **l)** no LTF scheme. Air parcels have been binned to 5°-longitude x 2°-latitude grid.

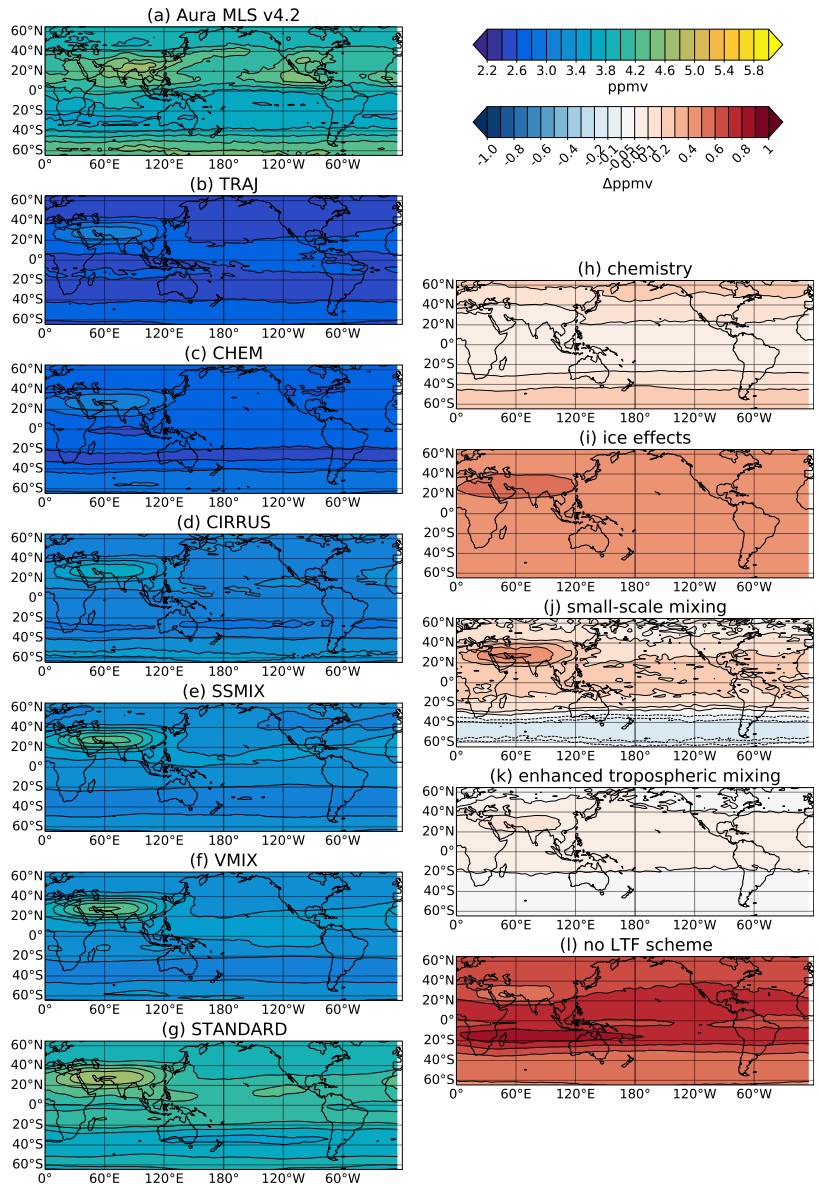

**Figure 2.** (Left column) Climatology of water vapour distribution at 80 hPa during boreal summer (JJA) for the period 2007-2016 from **(a)** Aura MLS v4.2 observations, **(b)** TRAJ, **(c)**CHEM, **(d)**CIRRUS, **(e)**VMIX, **(f)**SSMIX experiments and **(g)**STANDARD simulation. (Right column) Isolated effect of each **(h)**chemistry, **(i)** cirrus, **(j)** small-scale mixing, **(k)** enhanced tropospheric mixing and **(l)** no LTF scheme. Air parcels have been binned to 5°-longitude x 2°-latitude grid.

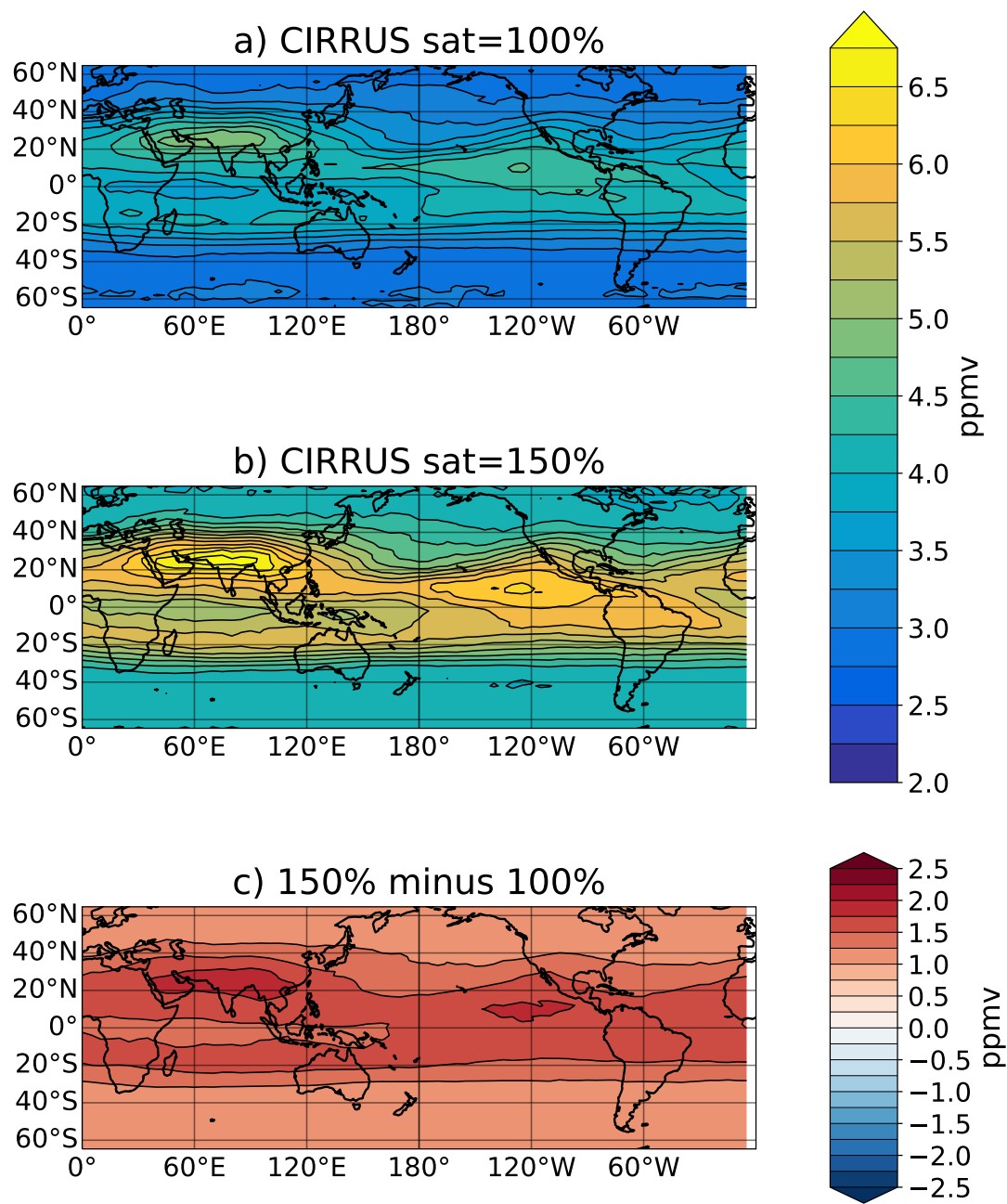

**Figure 3.** Distribution of water vapour at 100 hPa of CIRRUS using **(a)** 100% and **(b)** 150% as saturation mixing ratio respect to ice during boreal summer for 2005–2010.

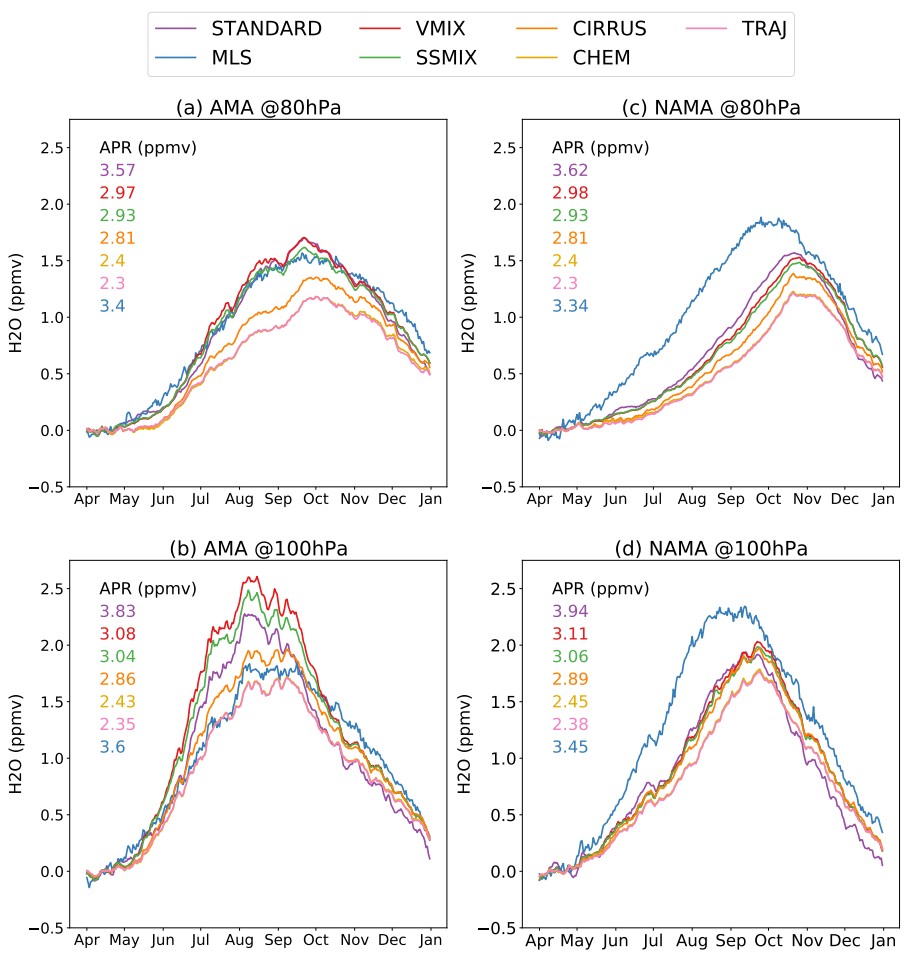

**Figure 4.** Amplitude of the cycle of daily water vapour at (top) 80 hPa and (bottom) 100 hPa averaged over **(a, b)** the Asian Monsoon Anticyclone, AMA (20°N–40°N, 40°E–140°E) and **(c, d)** the North American Monsoon Anticyclone, NAMA (10°N–30°N, 220°E–300°E) for the period 2007–2016. Colored numbers are the mean water vapour during April in each experiment, which is used as reference level. The regions in which averages are computed correspond to the maxima of water vapour found in the boreal summer climatology of the water vapour observed by Aura MLS v4.2.

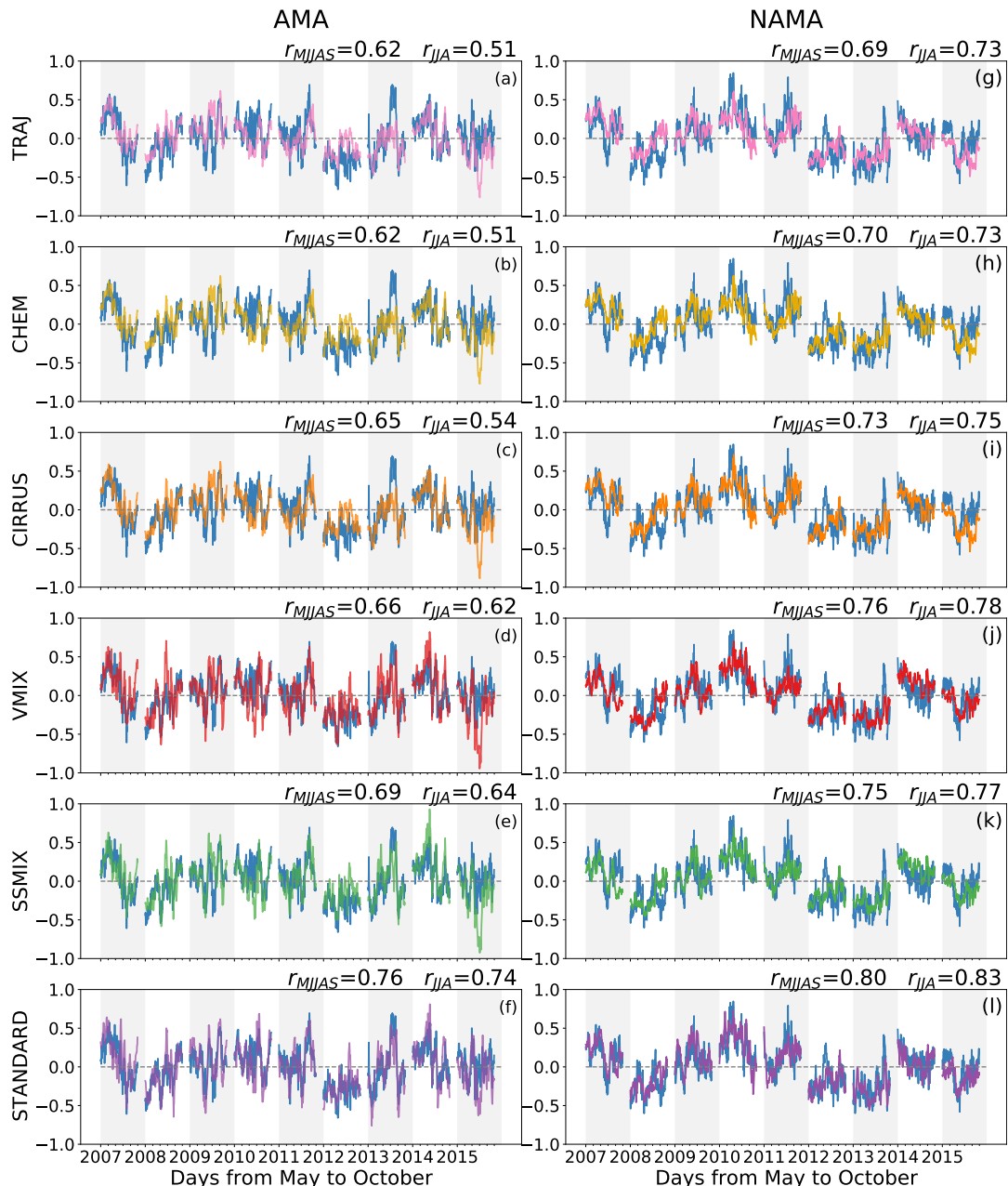

**Figure 5.** Boreal summer deseasonalized anomalies of water vapour in (left column) the Asian Monsoon Anticyclone, AMA, and (right column) the North American Monsoon Anticyclone, NAMA, for (top-bottom) TRAJ, CHEM, CIRRUS, VMIX, SSMIX, STANDARD in comparison with MLS (blue line). Correlation values between each experiment and MLS are calculated from May to Sep ($r_{MJJAS}$) and from June to August ($r_{JJA}$).

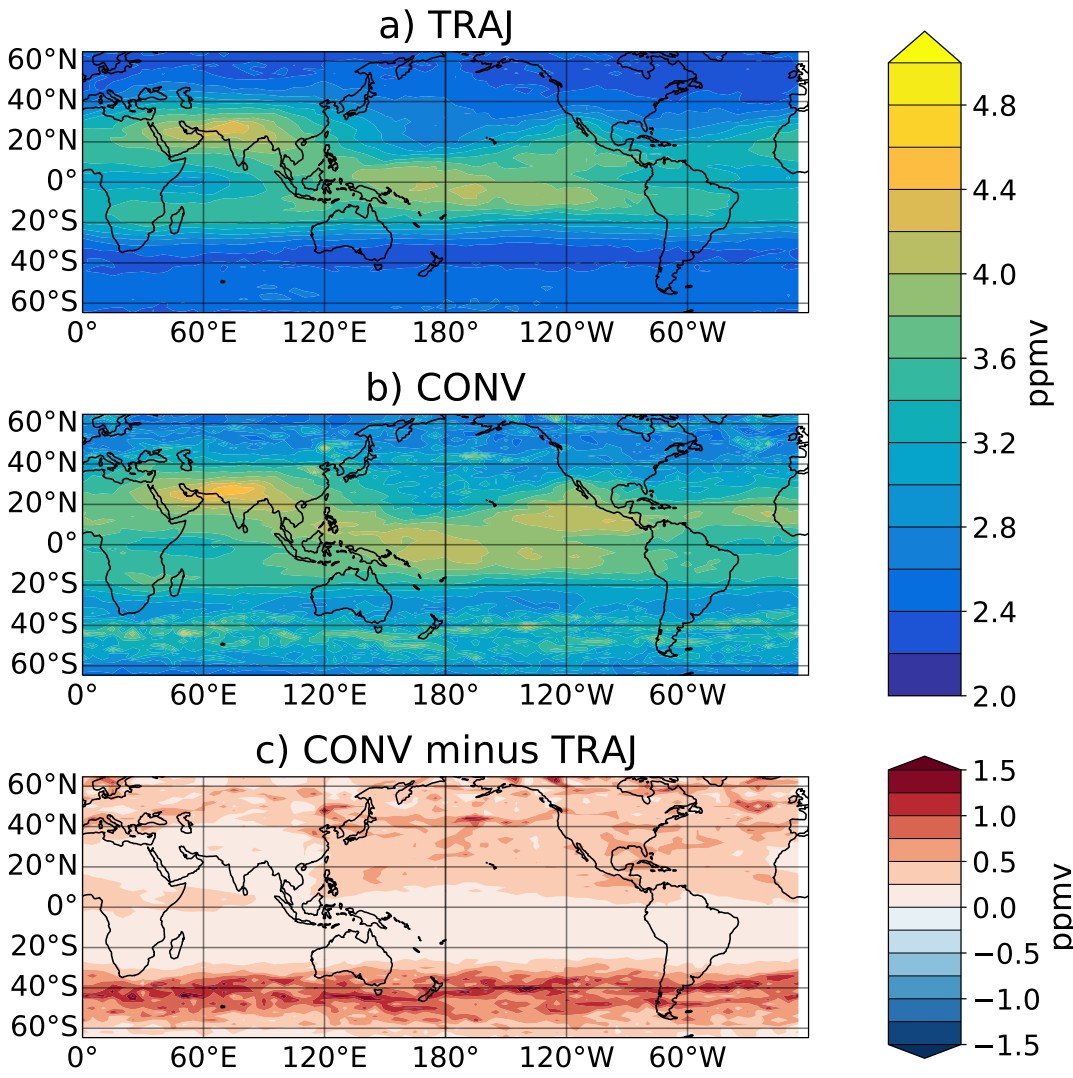

**Figure 6.** Boreal summer distribution of water vapour at 100hPa in 2008 for **(a)** TRAJ and **(b)** CONV experiment and **(c)** their differences

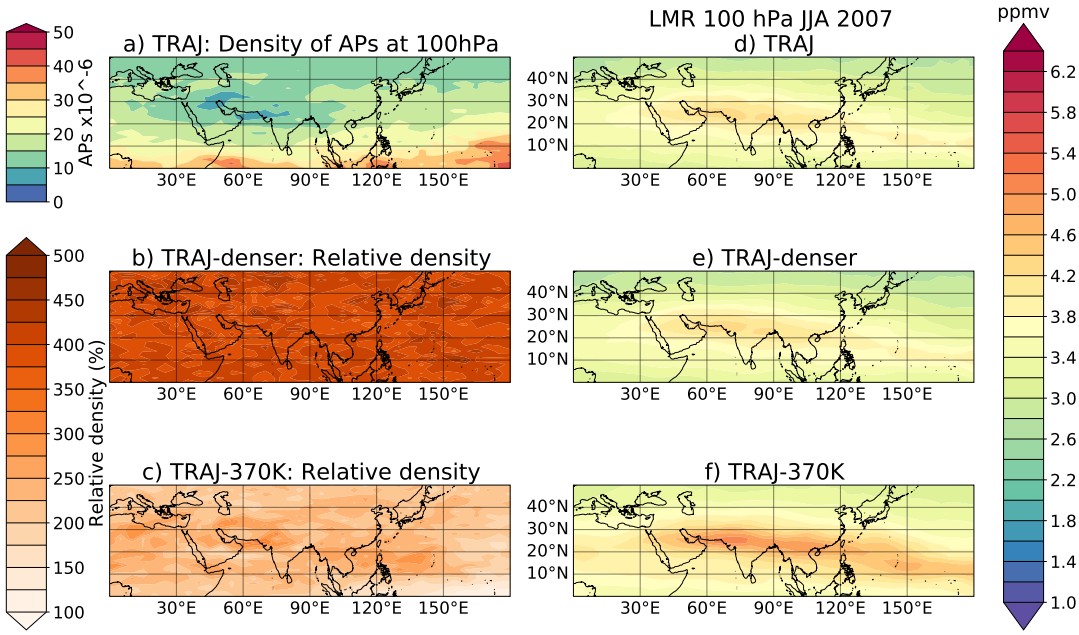

**Figure 7. (a)** Normalized density of air parcels in the AMA region during JJA in 2007 for TRAJ experiment. Distributions of the relative number of air parcels respect to TRAJ in **(b)** TRAJ-denser and **(c)** TRAJ-370 K. Boreal water vapour distributions of **(d)** TRAJ, **(e)** TRAJ-denser and **(d)** TRAJ-370K in 2007.

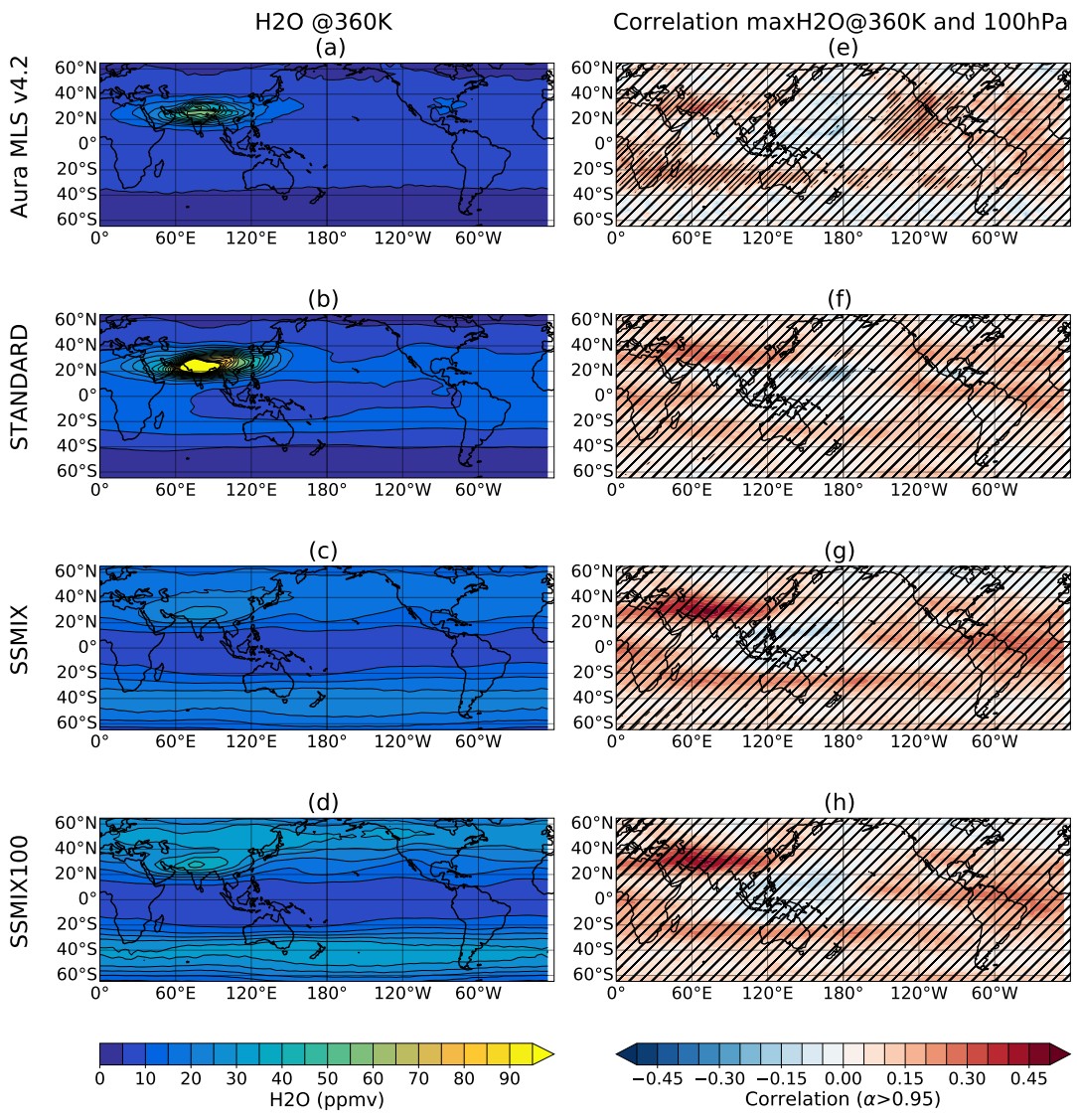

**Figure 8.** (Left column) Climatology of water vapour at 360 K from **(a)** Aura MLS v4.2 observations, **(b)** STANDARD, **(c)** SSMIX and **(d)** SSMIX initialized with 100 ppmv. (right column) Correlation between the water vapour timeseries averaged in 60°E–100°E, 20°N–30°N at 360 K and the timeseries of water vapour at 100 hPa at each grid point during boreal summer for **(e)** Aura MLS v4.2 **(f)** STANDARD, **(g)** SSMIX and **(h)** SSMIX initialized with 100 ppmv.

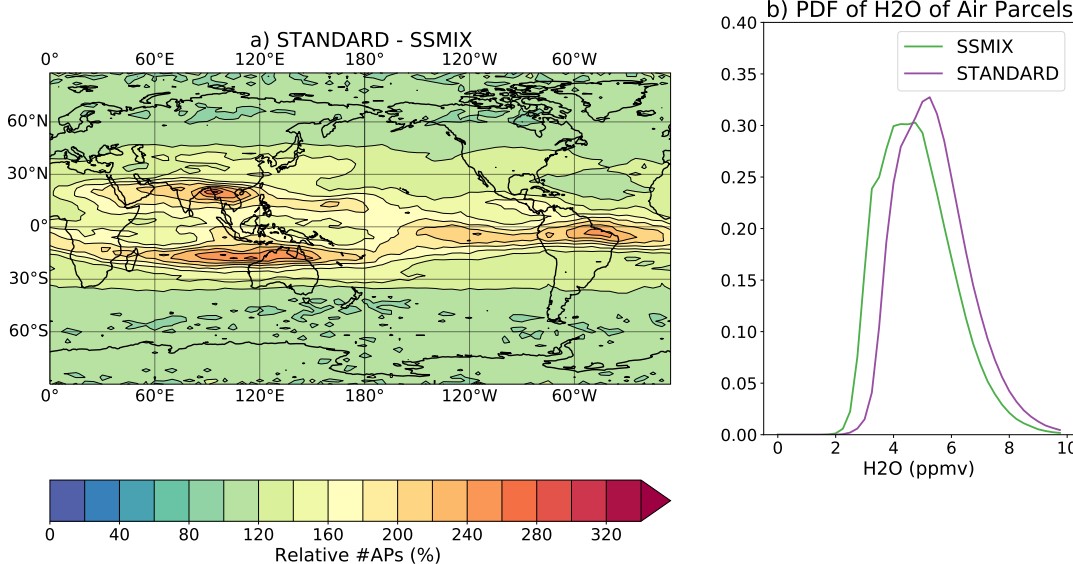

**Figure 9. (a)** Distribution of the relative number of air parcels simulated in STANDARD with respect to SSMIX during JJA for 2007–2016.
**(b)** Normalized PDF of the water vapour of air parcels encountered during JJA (2007–2016) over the Asian Monsoon region ($20°–40°N$, $40°–140°E$) for SSMIX and STANDARD.

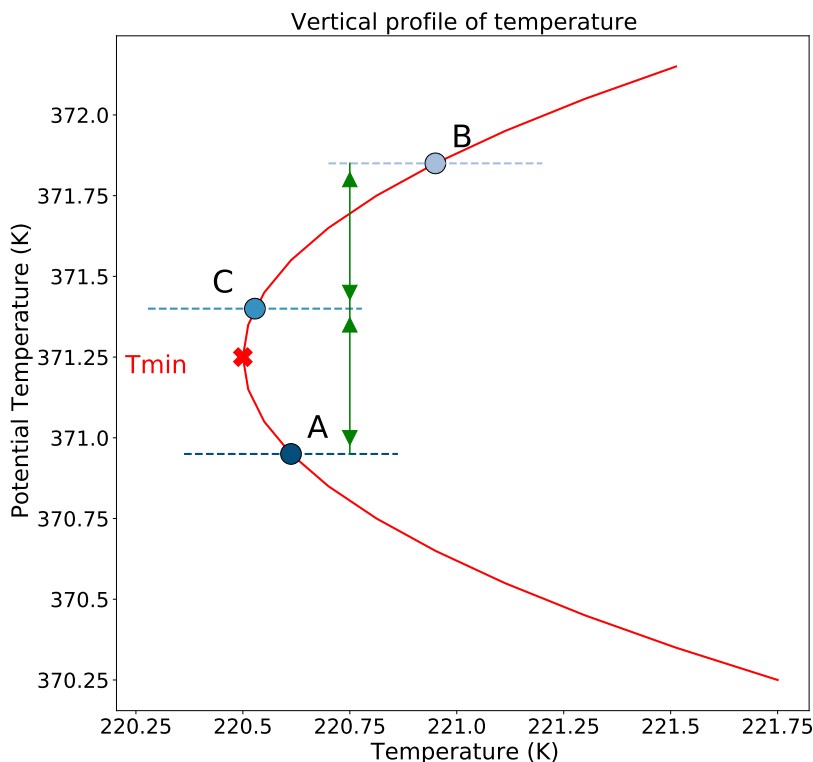

**Figure A1.** Hypothetical scheme in which the mixing process avoids a "cold trap". Air parcels A and B mix into C. Due to the temperature vertical profile, temperature in C is larger than minimum temperature registered below, Tmin. Therefore, in case C is saturated according to its temperature, the water vapour content would be larger than if an air parcel would be transported to the same altitude encountering Tmin.

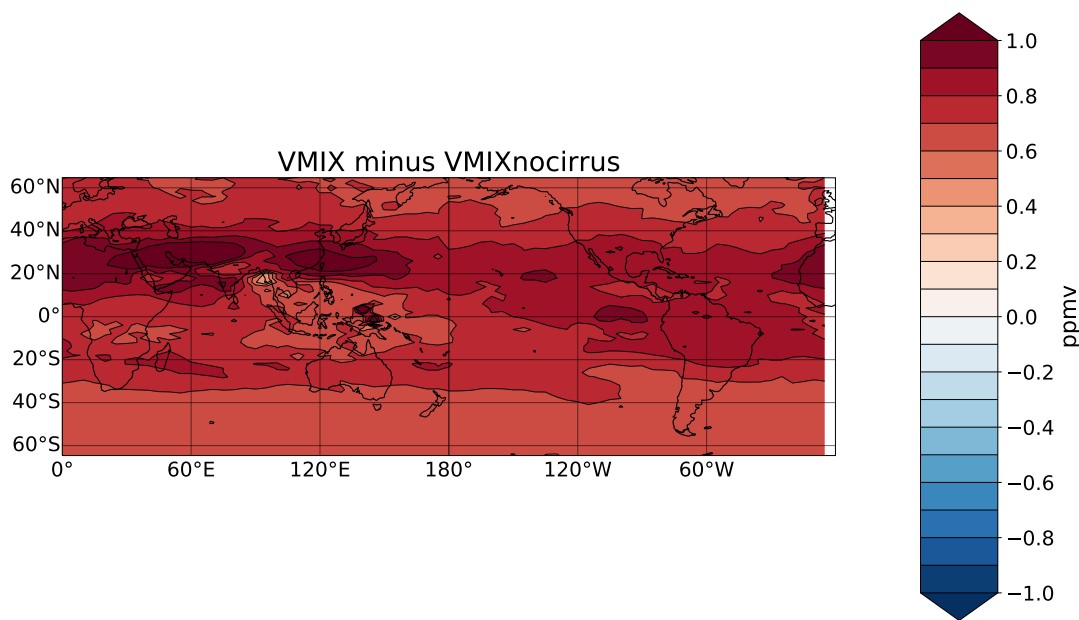

**Figure A2.** Differences in the distribution of water vapour at 100 hPa between VMIX (experiment with ice microphysics) and VMIXnocirrus (experiment without ice microphysics) during JJA for 2005–2008. Red colors mean VMIX performs larger water vapour than VMIXnocirrus.

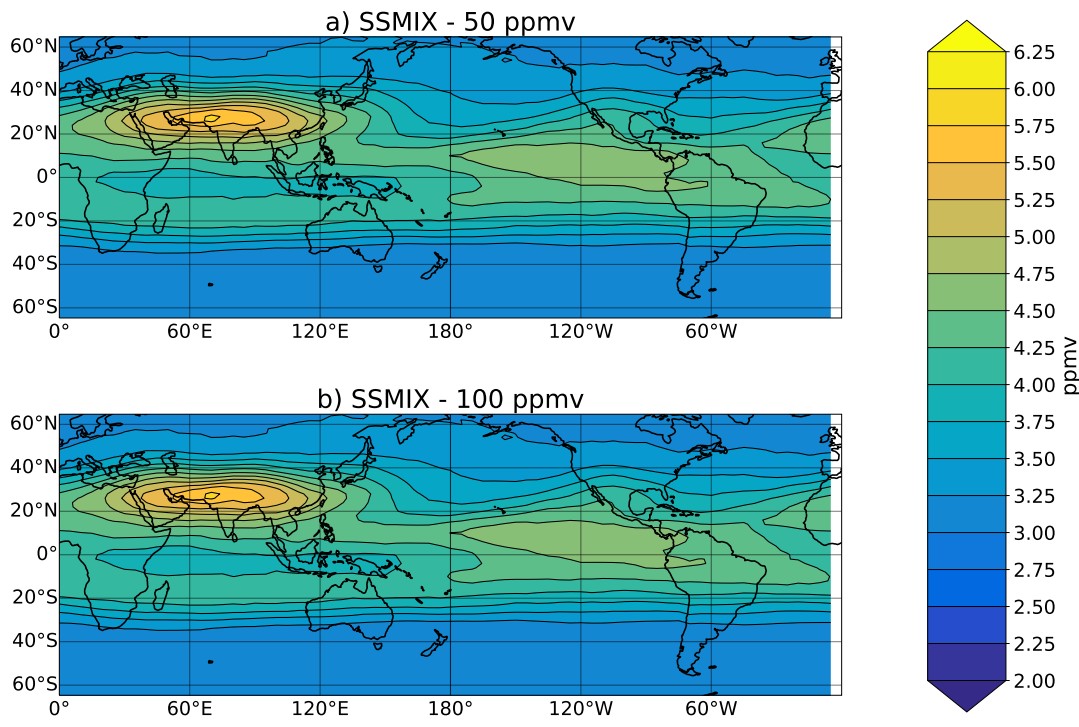

**Figure A3.** Distribution of water vapour at 100 hPa of SSMIX initialized with **(a)** 50 ppmv and **(b)** 100 ppmv during boreal summer for 2007–2016.

**Table 1.** Description of experiments done with CLaMS

| Experiment | Configuration | Timestep | H2O at 360K | #Air parcels | Further details |
|---|---|---|---|---|---|
| **TRAJ** | LTF | 6h | None | ∼412000 | Pure advective trajectories using ERAinterim horizontal wind fields and diabatic heating rate. |
| **CHEM** | TRAJ + chemistry module | 6h | 50 | ∼412000 | Only methane oxidation |
| **CIRRUS** | CHEM + cirrus scheme | 6h | 50 | ∼412000 | characteristic length set to ∼300m |
| **SSMIX** | CIRRUS + small-scale mixing | 24h | 50 | ∼1026000 | After mixing, cirrus scheme is applied again |
| **VMIX** | SSMIX + tropospheric mixing | 24h | 50 | ∼1026000 | After mixing, cirrus scheme is applied again |
| **STANDARD** | SSMIX + ST-Filling | 24h | ERAInterim | ∼2000000 | After mixing, cirrus scheme is applied again. Full chemistry (see McKenna et al. (2002a)) No LTF set up Water vapour fields from ERAInterim in troposphere below 500 hPa. |

*Timestep* specifies the frequency of the output in each experiment.

#Air parcels is the mean number of air parcels per day after 2-year of spin-up time

LTF *(Lagrangian Trajectory Filling Set Up)*: based in the domain-filling technique developed by Schoeberl and Dessler (2011)

ERAinterim reanalysis from European Centre for Medium Range Weather Forecasts