# Peer review of "Processes influencing lower stratospheric water vapour in monsoon anticyclones: insights from Lagrangian modeling"

_Atmospheric Chemistry and Physics, 2020_

## Referee Comment (RC1) · Anonymous Referee #1 · 17 Nov 2020

Overview: This paper uses the CLaMS trajectory model to fill the stratosphere with parcels which the authors call Lagrangian Trajectory Filling (LTF). This technique (when mixing is shut off) is identical to Forward Domain Filling (FDF) pioneered by Schoeberl and Dessler (2011). Using a series of experiments the authors investigate the water vapour anomalies associated with the monsoon regions comparing their results to those obtained by other authors using trajectory simulations.

Upper tropospheric and stratospheric water vapour can be tricky to simulate since convective sources, small scale temperature anomalies due to gravity wave and cloud formation processes all play a role - see Schoeberl et al. (2018) Fig. 3. CLaMS pa-

rameterization of mixing is unique and therefore brings an additional process to bear.

The microphysical model in this simulation is quite simple compared to the schemes used by Ueyama et al. and Schoeberl et al. This could be important in determining the water vapour field in the upper troposphere and the conclusions of the authors. Line 166 describes their scheme. Basically, water vapour in excess of saturation is made available for ice particle formation, since the particle number density is imposed, this fixes the particle size, and dehydration occurs through settling. Particle number densities are derived from Krämer et al (2009). First, rereading Krämer, it wasn't clear whether the particle number densities used here were temperature dependent as shown in Krämer Fig. 5. Second, with fixed particle sizes, this scheme will likely overestimate dehydration. Once crystals form, particle growth occurs and the dehydration rate is initially slow because the settling rate is slow for small particles. If the parcel warms up during the beginning of the cloud formation process, the ice will evapourate producing almost no dehydration. This is how short horizontal wavelength gravity waves can produce clouds with almost no effective dehydration. The author's formulation of the microphysics, I believe will low bias the water vapour concentration. Third, the saturation level (100%RH) is used to trigger dehydration yet Krämer clearly shows that UTLS air can be supersaturated without cloud formation (a result also found by ATTREX flights, Jensen et al., 2017). Neglect of super saturation will also low bias the water vapour compared to observations.

Another important consideration is the convective parameterization used in this simulation. Ueyama et al. (2018) used observed convective heights to add water to the parcel distribution if the parcels are below the convective top and near the convection. I am not exactly sure what is used here (this aspect of the paper needs improvement, but on lines 331, 388 it states that convection is not included). Convection is an important part of the water vapour budget over the monsoon regions, and not including it colors the validity of the simulations and conclusions reached here. Lack of convective moistening could also lead to the low water vapour bias over the monsoons shown in

the CIRRUS simulation, for example. I suggest that you take the approach used by Ueyama and Schoeberl. Get the convective heights from ERAi and saturate parcels nearby and below convective tops.

CLaMS apparently mixes water vapour as it mixes other tracers - as described in McKenna et al. (2002) - and that transport can be cross isentropic. Unlike chemical tracers, water vapour concentration can be temperature sensitive if saturation is reached. Does CLaMS consider that the mixing between parcels may undergo temperature excursions that could remove water due to ice formation? In strong shear zones, the Richardson number will fall below 1/4 and the turbulent field will produce strong temperature excursions, cloud decks and dehydration. In any event, Is the total water content mixed - ice plus water vapour or just the vapour?

The authors spend some time discussing how LTF might be biased by having too few parcels in the AMA anticyclone. They note that CLaMS mixing simulations - by spawning new parcels - resolves this problem. But by spawning new parcels, CLaMS increases the parcel density over the whole domain (Table 1). For a rational comparison, the authors should try in increase the LTF injection rate to improve resolution above AMA. If the water vapour field above the AMA begins to converge they likely have reached a high enough injection rate.

The authors make many comparisons to MLS, but they should run their model simulations through the MLS vertical averaging kernel to correctly make such comparisons. This will tend increase the water vapour in the models somewhat because of the strong non-linear vertical gradient in water in the upper troposphere.

Overall, I like the idea that CLaMS is introducing a new - and probably relevant process - into the discussion of water vapour and determining the impact over the monsoon. The lack of convective moistening provides us with no real insight on how important the mixing processes might be. Also, looking at Figure 2, it looks to me like mixing is adding too much water. Below I have made a large number of suggestions to improving this

paper and I hope the authors make appropriate revisions and resubmit the manuscript.

Minor comments: Ln 28 also reference Randel and Park (2019; JGR)

Ln 43 you may also want to reference Randel et al 2011 for a discussion of the differences between AMA and NAMA with regard to the water vapour field.

Ln 78 'has not been assessed yet' please see Schoeberl et al. (2018) Fig. 3

Ln 84 - what does 'they' refer to?

Ln 100 Please use the latest version of MLS. V4.2 is somewhat wetter than V5

Ln 138, 145 Mention here that small scale mixing by CLaMS spawns new parcels thus producing a large variation in # of parcels from 400,000 to 20 million shown in Table 1.

Ln 153 Assuming that LMR is defined by 100% RH? Please be specific.

Ln 168 Please elaborate on what the 'characteristic length' is? The loss of ice from a parcel per time step is $\sim$ Ice*ws*?t/L where Ice is the ice mixing ratio, ws is the settling velocity, ?t time step and L is the cloud depth. Is L the characteristic length?

Ln 185 Please explain how supersaturation can develop after the mixing step if you have already restricted supersaturation before mixing. Also in the small scale mixing, is the ice divided up as well ?

Ln 187 Does convective moistening also occur with the convective updrafts? Shouldn't you be carrying ice into the updraft region. It seems to me this could be important to the water vapour budget over the monsoons.

Ln 200 I don't understand what is going on in STANDARD. Parcels released at 360 ascend into the stratosphere, dehydrate, then descend into the troposphere and mix with other parcels. It seems to me this would produce a very dry troposphere compared to what is observed, if I am understanding this correctly.

Ln 210 It would be useful to see a distribution of parcels with altitude for the various experiments. The STANDARD experiment I expect would have a large number of parcels in the troposphere.

Ln 213 You should update to MLS V5

Ln 220 To make an exact comparison to MLS you should run the model output through the MLS averaging kernels. Please explain why you did not do this, or indicate that you did do this.

Ln 227 I am not surprised that Traj and Chem have such low water vapour as has been found by others (e.g. Schoeberl et al., 2016). Basically, the inclusion of a cloud model and setting the nucleation RH to greater than 100% increases water vapour substantially over simply using the LDP value of water.

Ln 240 None of this is surprising and consistent with the water vapour budget of the stratosphere. You could use a few references here on methane oxidation and conservation of 2 CH4+H2O in the stratosphere.

Ln 260 The fact that small scale mixing increases water mostly in the monsoon only is a puzzle. According to you the mixing avoids the cold traps, but adiabatic turbulence produce cold temperatures and dehydration? The mixing scheme transfers water but doesn't take into account the temperature variation during that transfer - thus it would always overestimate the moistening by mixing. Since the model lacks ice injection by convection we can't tell if this mixing process competes with convective moistening.

Ln 277 I am not sure I agree that the effect of convective updrafts are limited by removing air parcels below 250hPa. The authors need to explain in more detail how mixing enhanceds convection. The authors should also re-read how convective influence is parameterized in Ueyama et al. (2018). Ice is added to parcels passing near convection that are below the convective cloud tops. A similar scheme is used by Schoeberl et al. (2019). The advanced cloud model in Ueyama et al. hydrates the air appropriately for parcels that have collided with convection. From ln 277 to ln 280 is pretty

speculative.

Ln 290 The STANDARD experiment shows interesting results, but I am not sure I agree with its conclusions. The question that needs to be asked is where does water in the mid-troposphere come from? In the tropics, water vapour is detrained from convection moistening air that is descending from even higher levels. In the mid-latitudes, moist air also rises along frontal systems. I have no doubt that CLaMS can simulate the horizontal transport of water vapour, but the rehydration of parcels through convective processes is not clearly specified. If the LTF is set up correctly, and water vapour fields are initiated at the 360 K surface from observations, the results should be correct. The comments about the deficiencies of LTF are based on the idea that STANDARD is correct which needs to be demonstrated. This point is reinforced later in the paper (Fig. 5) that shows STANDARD produces anomalous water vapour fields compared to MLS especially under the AMA anticyclone.

Ln 295 I agree that NAMA looks closer to MLS observations in STANDARD. But what about the high water vapour fields south of the equator? They are as large as NAMA and are not apparent in the MLS observations. I would argue that STANDARD is a worse simulation than SSMIX.

Ln 300 It would be very useful to put the temperature cycle on Figure 2 - at least at the tropopause level and perhaps the saturation mixing ratio. This might be a nice quantitative measure of how much water vapour is being enhanced by CLaMS mixing.

Ln 315... I would argue that VMIX, SSMIX and STANDARD do the worse job compared to other simulations based on the peak to valley change seen in MLS at 100 hPa. Basically, if you remove the offsets and judge the annual cycle, the CLaMS mixing is creating too much water during the monsoon in Fig. 2. It might be interesting to plot then all normalizing by the April value.

Ln 350 The fact that STANDARD, SSMIX and VMIX produce too rapid a rise in water vapour over the monsoon suggests to me that the mixing rate is too high. Since it can

be tuned lower, you might try a sensitivity experiment where the Lyapunov trigger is increased.

Ln 401 I totally agree that convection is important as I have argued above. So, in these model simulations there is only one process that can transport additional water into the upper troposphere: mixing. No wonder you conclude it is important. The study is flawed unless you include convection and compare the results to mixing.

Ln 405 I would argue that you need to tell us more details about water vapour mixing to make sure the readers understand the process. The fact that you have to invoke the dehydration process before and after the mixing step suggest that it is somewhat complicated. How often after you mix does the second application of dehydration actually do something. That would be interesting to know.

Ln 415 see comment on 401

Ln 438 I am not surprised by the lower density of air parcels over the AMA anticyclone shown in Figure A2 and the results from Fig 4. The divergent flow will tend to exclude parcels from that region, and the only source of parcels will those rising up from the region below. LTF is, in some sense, a natural sampling system (as opposed to Reverse Domain Fill where the sampling density is chosen ahead of time). The fact that there are some empty bins suggests that the gridding - which is arbitrarily chosen by the authors - is too small or that the parcel release rate is too low. This hypothesis is supported by the lack of gaps in STANDARD which has more than 4 times more parcels than CIRRUS (Table 1). It would be interesting to re-run the CIRRUS experiment quadrupling the parcel release rate. If this experiment is run, and the results are changed then this suggests that the CIRRUS experiment is operating with too low an injection rate and the water vapour field has not converged.

Ln 464 'explains why the occurrence of these gaps gives rise to drier conditions' I am somewhat confused by this statement. I certainly agree that LTF not resolving the AMA through lack of parcels will sample bias the water vapour, but I understood that you ar
C7

gued earlier in the paper that it was water vapour mixing by CLaMS that was increasing the water vapour in the AMA. I think that the experiment suggested above might be able to sort this out. It is likely that transport of water vapour through mixing (whether correct or not) is increasing water vapour in the AMA and the consequential increase of parcels through CLaMS spawning of new parcels is improving the sampling. I might add that LTF simulations performed by Schoeberl and colleagues typically have over 2 million parcels in the stratosphere, a resolution similar to STANDARD. The discussion in paragraph 467 is along the lines of the statements above - basically STANDARD is more successful because of the larger number of parcels.

Ln 488 'SSMIX ... is set to 50 ppmv' yet Figure 5c shows much lower values and a variation. Is this due to air parcels dehydrating at this level after release? You might want to add an explanation here similar to the point made on line 499

Ln 490 Figure 5a 'STANDARD water vapor (sic) distribution agrees quite well with MLS' You're kidding, right? The next couple lines outline the quite large differences. In any event, it seems like some of the good agreement at 100 hPa due to STANDARD is that air rising through the monsoon has very high water vapour amounts.

Please also note the supplement to this comment:
https://acp.copernicus.org/preprints/acp-2020-1010/acp-2020-1010-RC1-supplement.pdf

---

## Referee Comment (RC2) · Anonymous Referee #2 · 1 Dec 2020

This paper is based on a Lagrangian model CLaMS and mainly use the domain filling strategy to explore the influence of several processes to lower stratospheric water vapor, including temperature, methane oxidation, ice microphysics, small scale mixing, and tropospheric mixing. The authors pay attention to the Asian and North American regions, as well as compare it with the influence globally. They compare the model simulation result by adding each process and consider horizontal distribution, seasonal cycle, and the intra-seasonal anomaly of lower stratospheric water vapor. A comparison between the standard CLaMS model and the Lagrangian Trajectory Filling (LTF) strategy also has been made, and the authors discuss why the LTF strategy is dry-biased. Overall, the scientific topic the authors are exploring is of great interest in the

field of stratospheric water vapor. However, based on the concern of the scientific rigor of the model design, I recommend a major revision before publishing this paper.

General comments:

One big issue is the influence of individual factors is calculated as the difference between model simulations with and without this factor. For example, using the difference between the water vapor mixing ratio in CIRRUS and CHEM to represent ice effects. However, this value may yield different model designs. E.g., if compare the difference between the experiment VMIX, and the experiment with the same setting as VMIX but do not include the ice microphysics, will the value be the same as the difference between CIRRUS and CHEM experiments? Since the authors are trying to compare the contribution from different factors, more reasonable experiment settings or sensitivity tests would be the comparing STANDARD experiment and the experiments removing individual processes.

The second issue is that the Stratosphere-Troposphere filling experiment simulates closer results to the observation, so it is not clear why all of the previous experiments exploring the influence from chemical and physical processes are based on LTF strategy, instead of directly based on ST-Filling. At least the authors should provide an explanation of why most of the conclusions in this paper are drawn based on LTF stead of ST-filling.

Another recommendation is to add a sensitivity test based on the supersaturation level to the experiments if possible. It may influence the estimation of the influence of ice microphysics. And may also explain the low biased LTF simulation result.

The convection information derived from the ERAi troposphere is not reliable. The grid in ERAi is too coarse to capture deep convections into the stratosphere. Lacking convection is another possible reason for the low biased LTF simulation result. It may also influence the estimation of the influence of ice microphysics.

The last major issue is about the sensitivity test of water vapor in LTF in section 4.2. The authors conclude that there are many empty bins, or gaps, over the humid regions, and result in underestimation of the water vapor mixing ratio in LTF experiments. The question is if these gaps could be avoided, for example, by initiating air parcels on a denser grid. It seems that having empty bins is an indicator of not enough air parcels.

In addition, there are some minor issues:

Line 7 - . . .water vapor in that region and including it in the model simulation. . . -Is 'that' means AMA region?

Line 153 - . . .the water vapor simulated by this experiment corresponds to the Lowest Mixing Ratio (LMR) encountered by each air parcels along its trajectory - Is it an initial h2o mixing ratio or an upper bound of mixing ratio along the path? E.g. the lowest mixing ratio of some parcels may be very high.

Line 164 - the third experiment, CIRRUS. . . - Is the supersaturation level still 100% when considering the ice micro-physics? 100% may not be a realistic level.

Figure A1 – how do you calculate the water vapor mixing ratio on 80 hPa in MLS? By interpolation?

---

## Author Comment (AC1) · 10 Mar 2021

**1 General comments**

We thank the Reviewer for the very thorough and detailed comments which will definitely help to substantially improve the paper. We see the critical, but also very constructive, tone in some of the comments, and we did some extensive work (including substantial extension of the methodology, several additional sensitivity simulations, significant text changes) and think we can finally address all of the comments very well. Below, the reviewer comments are in blue, our replies in black, indicating parts in the

manuscript with major changes. The major changes in the revised version are:

- Clearer description of the paper goals (effects of small-scale mixing transports on monsoon water vapour), and more appropriate relation of the used methodology to literature. In particular, we made it clearer that the focus is on comparing effects of different processes in sensitivity simulations (with all these based partly on simplifying assumptions) and not presenting a "best simulation case".

- Added study of effects of supersaturation using sensitivity simulations with varied nucleation barrier (150% vs. 100% relative humidity) and added a new Fig. 3.

- We included effects of convection in the study, as suggested by the Reviewers, and added a discussion of related sensitivities (new Fig. 6).

- An analysis of the robustness of the calculation of the process effect as difference, from new sensitivity simulation VMIX without ice microphysics.

- Improved discussion of air parcel density in the monsoon, based on new sensitivity calculations.

**2 Specific comments**

**#1:** The microphysical model in this simulation is quite simple compared to the schemes used by Ueyama et al. and Schoeberl et al. This could be important in determining the water vapour field in the upper troposphere and the conclusions of the authors. Line 166 describes their scheme. Basically, water vapour in excess of saturation is made available for ice particle formation, since the particle number density is imposed, this fixes the particle size, and dehydration occurs through settling. Particle number densities are derived from Krämer et al (2009). First, rereading Krämer,

it wasn't clear whether the particle number densities used here were temperature dependent as shown in Krämer Fig. 5. Second, with fixed particle sizes, this scheme will likely overestimate dehydration. Once crystals form, particle growth occurs and the dehydration rate is initially slow because the settling rate is slow for small particles. If the parcel warms up during the beginning of the cloud formation process, the ice will evaporate producing almost no dehydration. This is how short horizontal wavelength gravity waves can produce clouds with almost no effective dehydration. The author's formulation of the microphysics, I believe will low bias the water vapour concentration. Third, the saturation level (100%RH) is used to trigger dehydration yet Krämer clearly shows that UTLS air can be supersaturated without cloud formation (a result also found by ATTREX flights, Jensen et al., 2017). Neglect of super saturation will also low bias the water vapour compared to observations.

**Reply:**We agree with the reviewer that the used approach simplifies key microphysical processes and their interactions with dynamics. Indeed, there are many unknowns in the treatment of microphysical processes in the TTL and, in particular, their interactions with gravity waves (including supersaturation and nucleation delays, heterogeneous nucleation, nucleation quenching by gravity waves, interactions with sedimentation). However, our goal in this paper is to focus on the effects of large-scale temperature and dynamics and the additional effect of small-scale atmospheric mixing, whose impact on water vapor in the lower stratospheric monsoon anticyclones has not yet been considered thoroughly. Hence, we do not aim at a detailed microphysical process description, which is better treated elsewhere, but try to keep it as simple as possible to enable clear interpretation of transport effects. Nevertheless, the simplified representation of microphysical processes used here has been shown appropriate in previous works and is still considered as state-of-the-art for a sensitivity analysis like here (e.g., Fueglistaler et al., 2005; Schoeberl et al., 2011; 2012; 2013; Zhang et al. 2016, Poshyvailo et al. 2018, Wang et al. 2019). We will better explain the used simplifications and the scientific focus of our approach in the revised paper to make these points clearer (e.g., the used particle number density, which is here taken not depending on temperature, and will also improve the discussion of relations to published work including more detailed microphysics. Furthermore, to highlight the potential role of the observed supersaturation within clouds, we have performed a sensitivity analysis varying that parameter (see answer to Reviewer 2).

**#2:** Another important consideration is the convective parameterization used in this simulation. Ueyama et al. (2018) used observed convective heights to add water to the parcel distribution if the parcels are below the convective top and near the convection. I am not exactly sure what is used here (this aspect of the paper needs improvement, but on lines 331, 388 it states that convection is not included). Convection is an important part of the water vapour budget over the monsoon regions, and not including it colors the validity of the simulations and conclusions reached here. Lack of convective moistening could also lead to the low water vapour bias over the monsoons shown in C2the CIRRUS simulation, for example. I suggest that you take the approach used by Ueyama and Schoeberl. Get the convective heights from ERAi and saturate parcels nearby and below convective tops.

**Reply:**Convection was indeed not included in the previous simulations, but we agree with the reviewer that it is an essential process which should not be ignored here. Therefore, we have decided to carry out and include additional sensitivity simulations with convection in the revised paper. Since ERA interim fields at about $1°$ rely on a convective parameterization essentially tuned to match the tropospheric heat and water budget, it is likely poorly suited for TTL water vapor and cloudiness. In order to include an estimate of the convective impact in our study, we follow the reviewer's suggestion (and Ueyama et al., 2018; Tissier and Legras, 2016) and use infrared brightness temperature from geostationary satellites to infer the convective cloud top information. More precisely, we use the NOAA GridSat-B1 product (Knapp et al. 2010) and ERA interim temperature and pressure profile data to deduce cloud top altitudes using the method outlined Tissier and Legras (2016).

Figure 1 shows the water vapour distributions at 100hPa performed by our nonconvective experiment, TRAJ, (Fig. 1a), our convective experiment CONV (Fig. 1b) and the difference between both experiments for JJA in 2008. According to these results, convection has little impact on the Asian Monsoon. This disagreement with Ueyama et al. (2018) could be caused by the different diabatic heating rates used to transport air parcels vertically. In CONV, we use ERA-Interim total diabatic heating rates, while Ueyama et al. (2018) use Yang's radiative diabatic heating rates (Yang et al. 2010).

Another difference is the time period of the study. While we consider the whole summer of 2008, Ueyama et al. (2018) focused on a 7-day period in the middle of the summer in 2007. To see if this issue explains our disagreement with Ueyama et al. (2018), we have plotted the convective water vapour distributions for the same 7-day period. However, our results still differ from Ueyama et al. (2018). Our experiment TRAJ (no convection) performs a maxima of water vapour not seen in their results with the non-convective experiment. In addition, CONV stays very similar to TRAJ during the same period of time. This disagrees with the increase in water vapour registered by Ueyama et al. (2018) using their experiment with convection.

Though our results show differences to Ueyama et al. (2018), they lead to similar conclusions as other studies that have found a weak to negligible impact of convection (Randel et al. 2015, Wang et al. 2019). Nevertheless, our approach still has several limitations. For instance, we have not included additional ice formed during the convective processes, which has proven to have an impact on the LS water vapour budget, according to Wang et al. (2019). These discrepancies should be studied in a deeper future follow-up study, in which all the sub processes involved in convection are more carefully controlled. However, we think that also the simplified parameterization of the convective effect as we included in the revised manuscript (and as suggested by the Reviewer) adds substantial additional value to the manuscript. The effects of convection are mainly described in the revised manuscript on P14, L443-481.

**#3:** CLaMS apparently mixes water vapour as it mixes other tracers - as described

in McKenna et al. (2002) - and that transport can be cross isentropic. Unlike chemical tracers, water vapour concentration can be temperature sensitive if saturation is reached. Does CLaMS consider that the mixing between parcels may undergo temperature excursions that could remove water due to ice formation? In strong shear zones, the Richardson number will fall below 1/4 and the turbulent field will produce strong temperature excursions, cloud decks and dehydration. In any event, Is the total water content mixed - ice plus water vapour or just the vapour?

**Reply:** In a turbulent layer, adiabatic temperature excursions indeed occur, but they are transient and might not be long enough to lead to dehydration. It is not obvious how to represent them and this effect is hence not accounted for - Thanks for pointing to that. What is however included is the effect of mixing on both mean potential temperature and water vapor separately. Due to the vertical temperature variability between two mixed air parcels (Fig. 2a) and the convexity of the Clausius-Clapeyron function (Fig. 2b), mixing might lead to supersaturation even when both mixed air parcels are subsaturated. To handle this, dehydration in the cirrus module is run both before and after the mixing. The description of the representation of the small-scale mixing effect is improved in the revised version (e.g., P17, L535-541).

**#4:** The authors spend some time discussing how LTF might be biased by having too few parcels in the AMA anticyclone. They note that CLaMS mixing simulations - by spawning new parcels - resolves this problem. But by spawning new parcels, CLaMS increases the parcel density over the whole domain (Table 1). For a rational comparison, the authors should try to increase the LTF injection rate to improve resolution above AMA. If the water vapour field above the AMA begins to converge they likely have reached a high enough injection rate.

**Reply:** One of the aims of this paper is to study the effects on water vapour distribution of the different setups, LTF and ST-Filling, used with CLaMS. Though the LTF scheme has been used frequently in the literature, we are not aware of reporting of its lower density of air parcels in the AMA compared to other regions. The existence of gaps in

a 5x2 longitude x latitude grid reflects this feature.

One way to avoid the presence of gaps is to enlarge the size of the bins when gridding the air parcels. Doing this, we increase the number of air parcels per bin and discard possible empty places. Figure 3 shows the water vapour average in the AMA for the TRAJ experiment using different bin sizes, 5x2 and 10x4 lonxlat-grid, and directly the Lagrangian air parcels water vapour content without projecting them onto a regular grid. Water vapour averages in the monsoon do not show large differences between these different computations. Therefore, our results with TRAJ seem to be invariant to the existence of gaps end, hence, the density of air parcels.

**#5:** The authors make many comparisons to MLS, but they should run their model simulations through the MLS vertical averaging kernel to correctly make such comparisons. This will tend to increase the water vapour in the models somewhat because of the strong non-linear vertical gradient in water in the upper troposphere.

**Reply:** We did not apply the AK in the submitted version by purpose, as we're interested in the sensitivity of model results to certain processes, and we don't want to smear out the detailed effects. However, we agree with the Reviewer that it is necessary to estimate the potential effect of the AK on the presented comparison, and we do so in Fig. 4. At 100hPa there is of course some sensitivity to applying the AK, as the vertical gradient is rather large, at 82hPa there is not much of an effect left. Importantly, also at 100hPa the AK effect does not change the patterns in the distribution in the tropics and subtropics. Hence, neglecting the AK effect in this paper will not affect our conclusions strongly. We included a short related comment in the revised version on P8, L235.

[Figure]

**3 Technical corrections**

**Ln 28:** also reference Randel and Park (2019; JGR) **Reply:**We have done it in the paper, as suggested.

**Ln 43** you may also want to reference Randel et al 2011 for a discussion of the differences between AMA and NAMA with regard to the water vapour field. **Reply:** We think the Reviewer refers to Randel et al. (2012). We included this relevant paper in the introduction (P2 L45).

**Ln 78** 'has not been assessed yet' please see Schoeberl et al. (2018) Fig. 3 **Reply:** Schoeberl et al. (2018) performed several experiments to study the impact of tropical convection on stratospheric water vapour, concluding that convection has little effect. However, their results are focused on the analysis of changes in global stratospheric water vapour, with special attention to the winter season, without studying the effect on monsoon regions during boreal summer. More recently, Schoeberl et al. (2019) found that convection in the Asian Monsoon is tied to the highest RHi region consistent with Ueyama et al. (2018), which has been already mentioned in our paper (P3, L70). However, they do not focus in explaining this process but in the TTL boreal winter. Nevertheless, we have changed our sentence to "has not been fully assessed yet." (P3,L77).

**Ln 84** - what does 'they' refer to? **Reply:** We were referring to air masses. The text has been modified accordingly.

**Ln 100** Please use the latest version of MLS. V4.2 is somewhat wetter than V5 **Reply:** As stated by the Reviewer, the most recent version of MLS is v5 which has been proven to be 5-10% drier than v4.2 at stratospheric levels but with increased humidity in the tropics at 147hPa. However, as this version was released during the period of writing the paper we just keep v4.2. Also, our main conclusions concern the differences between the various sensitivity simulations including different processes and are

unaffected by the version of used MLS data.

**Ln 138, 145** Mention here that small scale mixing by CLaMS spawns new parcels thus producing a large variation in # of parcels from 400,000 to 20 million shown in Table 1. **Reply:** We have done the change proposed, but only in L145 as in L138 we are describing the Schoeberl et al. Domain Filling technique in general.

**Ln 153** Assuming that LMR is defined by 100% RH? Please be specific. **Reply:** We have changed the paper to make this point clearer (P6, L161)

**Ln 168** Please elaborate on what the 'characteristic length' is? The loss of ice from a parcel per time step is ∝ Ice*ws*?t/L where Ice is the ice mixing ratio, ws is the settling velocity, ?t time step and L is the cloud depth. Is L the characteristic length? **Reply:** We agree that we were not clear about this aspect in the submitted manuscript - thanks for pointing that out! The characteristic length parameterizes the ice fall-out in a simple way, just as suspected by the Reviewer. Hence, in each model time step $t$ a sedimentation length for a mean spherical ice particle (of mean radius, given the ice water content and an empirical particle density, see e.g., Ploeger et al., 2013) is calculated as s = w*t and compared to the characteristic length L. The ice loss during that time step is Ice*(s/L). This is explained in more detail in the revised version (P6, L78).

**Ln 185** Please explain how supersaturation can develop after the mixing step if you have already restricted supersaturation before mixing. Also in the small scale mixing, is the ice divided up as well ? **Reply:** Please, see major comment #3 above.

**Ln 187** Does convective moistening also occur with the convective updrafts? Shouldn't you be carrying ice into the updraft region. It seems to me this could be important to the water vapour budget over the monsoons. **Reply:** The vertical mixing procedure enhances mixing in the vertical coordinate. This is done in the same way as small-scale mixing works. Therefore, not only water vapour but also ice is mixed, which could result in ice injection at higher altitudes. Once vertical mixing is considered, we apply again

the cirrus parameterization, to set possible supersaturated air parcels to saturation conditions (RH=100%). We clarified the respective text parts in the manuscript (P7, L209).

**Ln 200** I don't understand what is going on in STANDARD. Parcels released at 360 ascend into the stratosphere, dehydrate, then descend into the troposphere and mix with other parcels. It seems to me this would produce a very dry troposphere compared to what is observed, if I am understanding this correctly. **Reply:** STANDARD is a long multiyear full-CTM CLaMS simulation with air parcels initialized throughout the domain at the beginning of the simulation and thereafter initialized in each time step only in the boundary layer (e.g., Pommrich et al., 2014). Therefore, it differs from the rest of experiments that use LTF set up. In STANDARD, air parcels are not released at 360K but fill the whole atmosphere considered by the model, including the troposphere. Whenever the air parcels move below about 500hPa (exactly, the boundary condition is at 250K hybrid potential temperature, the vertical coordinate in CLaMS), their water vapour content is set to the water vapour fields of ERAinterim.

**Ln 210** It would be useful to see a distribution of parcels with altitude for the various experiments. The STANDARD experiment I expect would have a large number of parcels in the troposphere. **Reply:** The vertical distribution of air parcels for one given day in the TRAJ, SSMIX and STANDARD experiments is shown in Fig. 5. While LTF experiments, such as TRAJ and SSMIX, do not have air parcels below 250hPa, STANDARD fills up the troposphere as well (as explained in the reply to the previous comments).

**Ln 213** You should update to MLS V5 **Reply:** See our reply above.

**Ln 220** To make an exact comparison to MLS you should run the model output through the MLS averaging kernels. Please explain why you did not do this, or indicate that you did do this. **Reply:** See above (major comment #5).

**Ln 227** I am not surprised that Traj and Chem have such low water vapour as has been found by others (e.g. Schoeberl et al., 2016). Basically, the inclusion of a cloud
model and setting the nucleation RH to greater than 100% increases water vapour substantially over simply using the LDP value of water. **Reply:** We have added "as expected" (P8, L249).

**Ln 240** None of this is surprising and consistent with the water vapour budget of the stratosphere. You could use a few references here on methane oxidation and conser- vation of 2 CH4+H2O in the stratosphere. **Reply:** We have added "as expected" (P9,L265), and also references concerning methane oxidation (eg., Randel et al., 1998).

**Ln 260** The fact that small scale mixing increases water mostly in the monsoon only is a puzzle. According to you the mixing avoids the cold traps, but adiabatic turbulence produce cold temperatures and dehydration? The mixing scheme transfers water but doesn't take into account the temperature variation during that transfer - thus it would always overestimate the moistening by mixing. Since the model lacks ice injection by convection we can't tell if this mixing process competes with convective moistening. **Reply:** The mixing scheme is considering both temperature and water vapor, as explained above (answer to major comments). Only small-scale transient temperature fluctuations in turbulent layers are neglected, but those are short-lived. We now mention those assumptions explicitly in the main text. Furthermore, we have included an estimate of the convection effect as suggested by the Reviewer and compare the mixing results to that (as explained above).

**Ln 277** I am not sure I agree that the effect of convective updrafts are limited by removing air parcels below 250hPa. The authors need to explain in more detail how mixing enhances convection. The authors should also re-read how convective influence is parameterized in Ueyama et al. (2018). Ice is added to parcels passing near convec- tion that are below the convective cloud tops. A similar scheme is used by Schoeberl et al. (2019). The advanced cloud model in Ueyama et al. hydrates the air appro- priately for parcels that have collided with convection. From ln 277 to ln 280 is pretty speculative. **Reply:** We agree that convective updrafts are important for vertical transport of trac-

ers, such as the water vapour, into the LS. In our experiments, we consider this process inside VMIX. However, as VMIX is configured using the LTF scheme, air parcels are removed below 250 hPa. What we meant here is that this removing of parcels likely limits the accuracy of VMIX to reproduce properly convective updrafts. Therefore, our results cannot be seen as the full picture of convection yet. As described already above, in the revised version we included another sensitivity experiment to account for the effects of convection using a similar approach as Ueyama et al. (2018).

**Ln 290** The STANDARD experiment shows interesting results, but I am not sure I agree with its conclusions. The question that needs to be asked is where does water in the mid-troposphere come from? In the tropics, water vapour is detrained from convection moistening air that is descending from even higher levels. In the mid-latitudes, moist air also rises along frontal systems. I have no doubt that CLaMS can simulate the horizontal transport of water vapour, but the rehydration of parcels through convective processes is not clearly specified. If the LTF is set up correctly, and water vapour fields are initiated at the 360 K surface from observations, the results should be correct. The comments about the deficiencies of LTF are based on the idea that STANDARD is correct which needs to be demonstrated. This point is reinforced later in the paper (Fig. 5) that shows STANDARD produces anomalous water vapour fields compared to MLS especially under the AMA anticyclone. **Reply:** In the troposphere, the air parcels of STANDARD are set to ERAinterim water vapour values. Of course STANDARD has its limitations (as noted by the Reviewer), but in general it is the closest experiment to MLS as it is able to reproduce not only the spatial pattern of water vapour but also its variability in monsoon regions. Besides, LTF is not initialized using observations of water vapour, but setting an homogeneous 50-ppmv field at 360K, which is less realistic than STANDARD at that level. In any case, our results are independent of the initial water vapour used to release air parcels, as the comparison between the two sensitivity SSMIX experiments with different initial conditions (SSMIX-50ppmv and SSMIX-100ppmv) shows.

**Ln 295** I agree that NAMA looks closer to MLS observations in STANDARD. But what about the high water vapour fields south of the equator? They are as large as NAMA and are not apparent in the MLS observations. I would argue that STANDARD is a worse simulation than SSMIX. **Reply:** Thanks for this remark! It is true that SSMIX represents water vapour patterns in some regions better than STANDARD, but STANDARD simulates better the water vapour variability in those regions (correlations above 0.7 in AMA and NAMA compared to 0.6 achieved by rest of experiments and Zhang et al. (2017)). It is worth mentioning here, that the partial high-bias of STANDARD water vapour at 100hPa (Fig. 1) does not occur at upper levels (e.g., 82hPa). But we agree, it is indeed not straightforward whether STANDARD or SSMIX globally yield the better agreement with MLS. Hence, we changed the manuscript text accordingly to avoid a clear judgement of the best agreement, but focus more on the relative differences between the experiments related to the particular processes.

**Ln 300** It would be very useful to put the temperature cycle on Figure 2 - at least at the tropopause level and perhaps the saturation mixing ratio. This might be a nice quantitative measure of how much water vapour is being enhanced by CLaMS mixing. **Reply:** We do not think that considering mean tropopause temperatures will give much more insights. The water vapour at 100hPa is not set by the averaged tropopause temperature but by the Lagrangian Cold Point LCP (e.g., Zhang et al., 2017). However, the mean saturation mixing ratio at the LCP is just the water vapour in the TRAJ experiment in Fig. 4. Hence, comparison to TRAJ in Fig. 4 already provides the requested information.

**Ln 315** ... I would argue that VMIX, SSMIX and STANDARD do the worse job compared to other simulations based on the peak to valley change seen in MLS at 100 hPa. Basically, if you remove the offsets and judge the annual cycle, the CLaMS mixing is creating too much water during the monsoon in Fig. 2. It might be interesting to plot then all normalizing by the April value. **Reply:** Thanks for this good remark! Following the suggestions of the referee, we have replaced Fig. 2 by the amplitude of

water vapour cycles in the AMA for each experiment. Indeed, at 100hPa the annual cycle amplitude is too large in the mixing experiments when being compared to MLS. However, at 80hPa the amplitude in the mixing experiments is in very good agreement with MLS. The 100hPa level is frequently below the tropopause in the monsoon region, and the water vapour at this level can not yet be regarded as stratospheric entry values, but is very sensitive to small biases in tropopause height, etc. Hence, we think that comparison at slightly higher levels (here 80hPa) provides actually a better picture of which processes influence stratospheric entry water vapor. We tried to improve the related discussion in the manuscript (P11, L350).

**Ln 350** The fact that STANDARD, SSMIX and VMIX produce too rapid a rise in water vapour over the monsoon suggests to me that the mixing rate is too high. Since it can be tuned lower, you might try a sensitivity experiment where the Lyapunov trigger is increased. **Reply:** The proposed sensitivity study has been done in Poshyvailo et al. (2018) (their Figure). Here we used the optimized value they propose. And as described in the previous reply, at levels entirely above the tropopause (e.g., 80 hPa) this choice leads to the best agreement with MLS.

**Ln 401** I totally agree that convection is important as I have argued above. So, in these model simulations there is only one process that can transport additional water into the upper troposphere: mixing. No wonder you conclude it is important. The study is flawed unless you include convection and compare the results to mixing. **Reply:** See our response to Major comment #2.

**Ln 405** I would argue that you need to tell us more details about water vapour mixing to make sure the readers understand the process. The fact that you have to invoke the dehydration process before and after the mixing step suggest that it is somewhat complicated. How often after you mix does the second application of dehydration actually do something. That would be interesting to know. **Reply:** Please, see major comment #1 from Referee 2, in particular Figure 1.

**Ln 415** see comment on 401

**Ln 438** I am not surprised by the lower density of air parcels over the AMA anticyclone shown in Figure A2 and the results from Fig 4. The divergent flow will tend to exclude parcels from that region, and the only source of parcels will those rising up from the region below. LTF is, in some sense, a natural sampling system (as opposed to Reverse Domain Fill where the sampling density is chosen ahead of time). The fact that there are some empty bins suggests that the gridding - which is arbitrarily chosen by the authors - is too small or that the parcel release rate is too low. This hypothesis is supported by the lack of gaps in STANDARD which has more than 4 times more parcels than CIRRUS (Table 1). It would be interesting to re-run the CIRRUS experiment quadrupling the parcel release rate. If this experiment is run, and the results are changed then this suggests that the CIRRUS experiment is operating with too low an injection rate and the water vapour field has not converged. **Reply:** See comment #4 about the density of air parcels.

**Ln 464** 'explains why the occurrence of these gaps gives rise to drier conditions' I am somewhat confused by this statement. I certainly agree that LTF not resolving the AMA through lack of parcels will sample bias the water vapour, but I understood that you argued earlier in the paper that it was water vapour mixing by CLaMS that was increasing the water vapour in the AMA. I think that the experiment suggested above might be able to sort this out. It is likely that transport of water vapour through mixing (whether cor- rect or not) is increasing water vapour in the AMA and the consequential increase of parcels through CLaMS spawning of new parcels is improving the sampling. I might add that LTF simulations performed by Schoeberl and colleagues typically have over 2 million parcels in the stratosphere, a resolution similar to STANDARD. The discussion in paragraph 467 is along the lines of the statements above - basically STANDARD is more successful because of the larger number of parcels. **Reply:** Please, see response to comment #4 above. Regarding the number of air parcels in similar published studies, the number of parcels in our pure trajectory simulations (e.g., TRAJ) should

be similar, at least to Schoeberl et al. (2011), who state 500 000 parcels in their sim-
ulations (their paper, p. 8435). Therefore, we think that the set-up we use should be
comparable, at least with some, published experiments.

**Ln 488** 'SSMIX ... is set to 50 ppmv' yet Figure 5c shows much lower values and a
variation. Is this due to air parcels dehydrating at this level after release? You might
want to add an explanation here similar to the point made on line 499 **Reply:** Once air
parcels are released, they dehydrate according to the temperature of the region where
they have been launched. This is the main reason why the distribution of water vapour
at 360K does not correspond to a homogeneous 50ppmv-field. Besides, there are air
parcels coming from other levels that contribute to dehydrate the levels. Because of the
Brewer-Dobson circulation, most of the air parcels reaching this potential temperature
level come from high latitudes after leaving the stratosphere. This means that their
water vapour content is the characteristic of the stratosphere which is much lower than
50ppmv. As result, the horizontal distribution is an average between the downwarding
and spawning air parcels with the ones newly released.

**Ln 490** Figure 5a 'STANDARD water vapor (sic) distribution agrees quite well with MLS'
You're kidding, right? The next couple lines outline the quite large differences. In any
event, it seems like some of the good agreement at 100 hPa due to STANDARD is that
air rising through the monsoon has very high water vapour amounts. **Reply:** We agree
that our wording here was too positive and misleading. And actually we did not expect
a good quantitative agreement at levels below the tropopause (360K in the monsoon
region, where the tropopause can be as high as 400K), as the freeze-out process has
not been completed for many air parcels. We have deleted our comment.

[Figure]

**Fig. 1.** Boreal summer distribution of water vapour at 100hPa in 2008 for a) TRAJ and b) CONV experiment and c) their differences.

[Figure]

**Fig. 2.** Supersaturation caused by mixing due to (a) the occurrence of a local temperature minimum between the two mixed APs and (b) the convexity of the Clausius-Clapeyron relation.

[Figure]

[Figure]

**Fig. 3.** Water vapour averages in the AMA performed by TRAJ using (red) non-gridded air parcels, (green) 5°x2° and (blue)10°x4° <lonxlat> binned air parcels at 100 hPa.

**a)**

CLaMS, JJA (2004–2013), 0082hPa

**b)**

CLaMS, JJA (2004–2013), 0082hPa

**c)**

CLaMS, JJA (2004–2013), 0100hPa

**d)**

CLaMS, JJA (2004–2013), 0100hPa

**Fig. 4.** Distribution of water vapour simulated by STANDARD (left column) using the averaging kernels of MLS and (right column) not using them at a-b) 82 hPa and c-d) 100hPa.

[Figure]

**Fig. 5.** Vertical distribution of air parcels present in simulation the 8th of July 2013 for a) TRAJ,
b) SSMIX and c) STANDARD experiments.

---

## Author Comment (AC2) · 10 Mar 2021

**1 General comments**

We thank the Reviewer for the very thorough and detailed comments which will definitely help to substantially improve the paper. We see the critical, but also very constructive, tone in some of the comments, and we did some extensive work (including substantial extension of the methodology, several additional sensitivity simulations, significant text changes) and think we can finally address all of the comments very well. Below, the reviewer comments are in blue, our replies in black, indicating parts in the

manuscript with major changes. The major changes in the revised version are:

- Clearer description of the paper goals (effects of small-scale mixing transports on monsoon water vapour), and more appropriate relation of the used methodology to literature. In particular, we made it clearer that the focus is on comparing effects of different processes in sensitivity simulations (with all these based partly on simplifying assumptions) and not presenting a "best simulation case".

- Added study of effects of supersaturation using sensitivity simulations with varied nucleation barrier (150% vs. 100% relative humidity) and added a new Fig. 3.

- We included effects of convection in the study, as suggested by the Reviewers, and added a discussion of related sensitivities (new Fig. 6).

- An analysis of the robustness of the calculation of the process effect as difference, from new sensitivity simulation VMIX without ice microphysics.

- Improved discussion of air parcel density in the monsoon, based on new sensitivity calculations.

**2  Specific comments**

**#1:** One big issue is the influence of individual factors is calculated as the difference between model simulations with and without this factor. For example, using the difference between the water vapor mixing ratio in CIRRUS and CHEM to represent ice effects. However, this value may yield different model designs. E.g., if compare the difference between the experiment VMIX, and the experiment with the same setting as VMIX but do not include the ice microphysics, will the value be the same as the difference between CIRRUS and CHEM experiments? Since the authors are trying to compare the

contribution from different factors, more reasonable experiment settings or sensitivity tests would be the comparing STANDARD experiment and the experiments removing individual processes.

**Reply:** This is indeed a good comment, Thanks! First, we agree it would also be worth considering STANDARD as the reference and differencing from that case. However, as there is a number of existing studies based on pure trajectory approaches, we think that estimating the different effects as additions to such a set-up as reference (here TRAJ) is most useful for other groups.

Second, estimating the effect of ice microphysics is indeed somewhat tricky. In those experiments in which mixing is applied, here SSMIX and VMIX, we must consider the microphysics of ice after the mixing step. The reason behind is that mixing could lead to supersaturation of the air parcel. This situation is explained in Fig. 2b as response to comment #3 from Referee 1.

Considering two air parcels, "A" and "B", that are at the same pressure level and close enough to be mixed into C, the water vapour content of C would be the mean water vapour of A and B. However, the water vapour content of C could be larger (DH2O) than the corresponding 100% saturation conditions (see Fig. 2 in Reply to Reviewer 1. Therefore, mixing has produced a supersaturated air parcel C. With the second call of the microphysics scheme (after the mixing), the water vapour of C is set to saturation, and ice is formed from the water vapour in excess.

We understand the Reviewer's comment that this second application of the microphysics scheme could lead to a higher impact of ice on water vapour in "mixing" experiments. Following her/his suggestions, we have run a new experiment, here called "VMIXnocirrus", with the same setup as VMIX, but without the microphysics of ice. In this new experiment, whenever the water vapour content of the air parcel is above 100%relative humidity, all the water vapour in excess is removed (as in TRAJ). As in the case of "mixing experiments", SSMIX and VMIX, this process is applied twice, before

and after small-scale mixing, in VMIXnocirrus.

Figure 1 shows the isolated effects of ice in water vapour at 100hPa between CIRRUS and CHEM (Fig. 1a) and between VMIX and VMIXnocirrus(Fig. 1b). In Fig. 1a the ice microphysics scheme has been applied only once, while in Fig. 1b the scheme has been applied twice, before and after mixing. Clearly, the estimated ice effect increases for the second case (in simulations including mixing), due to calling the scheme two times. However, this increase is spatially rather homogeneous. There are some regions with a slightly stronger water vapour signal, such as the western flank of the Asian Monsoon and the Northwestern Pacific. As these regions are characterized by strong mixing, we think that it is the regional pattern in mixing causing these weak regional structures. (in Fig. 2a as response to comment #3 from Referee 1, we show a schematic situation in which mixing could "help" air parcels to avoid cold minima of the temperature's vertical profile).

In summary, the particular way of differencing to calculate the ice microphysics effect changes the estimated global value (by about 0.2 ppmv, see Fig. 1c), but not much the regional patterns (e.g., moisture anomaly in the monsoon). We note this in the revised version on P10, L295.

**#2:** The second issue is that the Stratosphere-Troposphere filling experiment simulates closer results to the observation, so it is not clear why all of the previous experiments exploring the influence from chemical and physical processes are based on LTF strategy, instead of directly based on ST-Filling. At least the authors should provide an explanation of why most of the conclusions in this paper are drawn based on LTF stead of ST-filling

**Reply:** As mentioned already in the response to the previous comment, pure back trajectory (LTF) approaches are very frequently used to study UTLS water vapour. In fact, several studies have used the domain filling technique developed by Schoerbel et. al (2011), here called "LTF" scheme, to study water vapour in the Lower Stratosphere.

Some of the most recent papers are Wang et al. (2019), Schoeberl et al. (2018), Zhang et al. (2016). In those papers, some processes are considered to influence water vapour, apart from freeze-out at the Lagrangian Cold Point. We regard our results most useful if they can be directly applied to those studies, and hence we think it is advantageous to use the pure trajectory approach TRAJ as baseline. In particular, it has never been studied before, how the LTF setup could influence the water vapour results. The comparison between the LTF scheme experiments and the CLaMS full-CTM simulation (STANDARD) here allows addressing this issue. Following the suggestion from the reviewer, we have clarified this in the paper (e.g., P4, L97).

**#3:** Another recommendation is to add a sensitivity test based on the supersaturation level to the experiments if possible. It may influence the estimation of the influence of ice microphysics. And may also explain the low biased LTF simulation result.

**Reply:** We agree that this would be a very valuable addition. We have performed a second run of CIRRUS but setting the critical RH barrier for ice nucleation to 150% instead of 100%. Figure 2 shows that considering a higher RH value amplifies the pattern of water vapour at 100hPa, especially in those regions in which ice microphysics are expected to have a stronger impact (e.g., Asian monsoon). As less ice is formed, air parcels can transport more water vapour upward. Besides, a higher critical RH value leads to less water vapour in excess and smaller ice particles. Then, slower sedimentation of the ice particles prevents them from being removed from simulation (in the CIRRUS scheme a sedimentation length is calculated assuming mean spherical ice particles, as explained also in the reply to Reviewer 1 (specific comment L168), and in the revised manuscript (P11, L327). In other words, a larger reservoir of water vapour for future air parcel's excursions into subsaturated regions is available in RH=150% than in RH=100%. case We have included this sensitivity test and a new figure (Fig. 3) in a new section in the discussion in the revised manuscript.

**#4:** The convection information derived from the ERAi troposphere is not reliable. The grid in ERAi is too coarse to capture deep convections into the stratosphere. Lacking

convection is another possible reason for the low biased LTF simulation result. It may also influence the estimation of the influence of ice microphysics.

**Reply:** As suggested by Reviewer 1 (see major comment #2 there), we have now included a simulation in which the effect of convection is estimated from observed convective cloud top information. We added these results to the revised manuscript to quantify the impact of convection on water vapor in the AMA. The effects of convection are mainly described in the revised manuscript as a new subsection in the Discussion, along with a Figure with our results (P14, L443-481, Fig.3). For further details see also our reply to Reviewer 1 (major comment #2).

**#5:** The last major issue is about the sensitivity test of water vapor in LTF in section 4.2. The authors conclude that there are many empty bins, or gaps, over the humid regions, and result in underestimation of the water vapor mixing ratio in LTF experiments. The question is if these gaps could be avoided, for example, by initiating air parcels on a denser grid. It seems that having empty bins is an indicator of not enough air parcels.

**Reply:** A similar comment was formulated by Reviewer 1. See our reply to Rev. 1 (major comment # 4).

**3   Technical corrections**

**Ln 7 -** . . .water vapor in that region and including it in the model simulation. . . -Is 'that' means AMA region?

**Reply:** Done. We have changed "that" to AMA to clarify it.

**Ln 153** . . .the water vapor simulated by this experiment corresponds to the Lowest Mixing Ratio (LMR) encountered by each air parcels along its trajectory - Is it an initial h2o mixing ratio or an upper bound of mixing ratio along the path? E.g. the lowest mixing ratio of some parcels may be very high.

**Reply:** In this experiment, the saturation mixing ratio of each air parcel is computed using Murphy and Koop (2005)'s formula. If during a calculation timestep the air parcel experiences a saturation mixing ratio lower than the actual water vapour mixing ratio, then its actual water vapour value is set to the saturation value (same as in many previous trajectory approaches, e.g., Fueglistaler et al., 2005: Schoeberl et al., 2011). Otherwise, if an air parcel shows mixing ratios higher than the LMR already experienced before, then its mixing ratio is not updated and the LMR remains the same. Therefore, the initial LMR of the air parcels released in TRAJ is the Mixing Ratio computed following Murphy and Koop (2005) which only depends on the temperature experienced by the air parcel, which is interpolated from the reanalysis field, and its pressure. However, we carried out an additional sensitivity test (changing the initial water vapour mixing ratio) to show that final water vapour value is largely independent from the initial value (P16, L488).

**Ln 164** the third experiment, CIRRUS. . . - Is the supersaturation level still 100% when considering the ice micro-physics? 100% may not be a realistic level. **Reply:** Schoeberl et al. (2018) performed several experiments to study the impact of tropical convection on stratospheric water vapour, concluding that convection has little effect. However, their results are focused on the analysis of changes in global stratospheric water vapour, with special attention to the winter season, without studying the effect on monsoon regions during boreal summer. More recently, Schoeberl et al. (2019) found that convection in the Asian Monsoon is tied to the highest RHi region consistent with Ueyama et al. (2018), which has been already mentioned in our paper (P3, L70). However, they do not focus in explaining this process but in the TTL boreal winter. Nevertheless, we have changed our sentence to "has not been fully assessed yet." (P3,L77).

**Figure A1** how do you calculate the water vapor mixing ratio on 80 hPa in MLS? By interpolation?

**Reply:** No, the level of MLS used is 82hPa which is very close to 80hPa. We have

specified this issue in the Methodology.

[Figure]

[Figure]

**Fig. 1.** Isolated ice effects computed as the difference in water vapour between (a) CIRRUS and CHEM, (b) VMIX and VMIXnocirrus and (c) the differences between both distributions at 100hPa during 2005-2007

[Figure]

**Fig. 2.** Boreal summer distribution of water vapour of CIRRUS experiment using (a) RH=100% and (b) RH=150% during 2005-2008 and (c) their differences.

---

## Author Response (AR2)

We thank the Reviewer for the very suitable comment made to our paper. The analysis derived from the comment has substantially improved both paper and the understanding of the scientific question. The major changes in the revised version are:

- Added a paragraph in the Discussion section, where we compared the results of two new experiments against former TRAJ experiment, as part of the suggested sensitivity study proposed by the Reviewer. The two new experiments consist either in a denser initialization grid or a higher initial potential temperature to release the daily new air parcels.
- Added a figure with the corresponding results from the sensitivity analysis.